# CameraNoise: Enabling Faithful Camera Control in Video Diffusion through Geometry-Flow-Guided Noise Warping

Haoyu Zhao [1 2 ♠]   Jiaxi Gu [3 ◇]   Haoran Chen [1 2]   Qingping Zheng [4]   Yeyin Jin [3]   Hongyi Yang [1 2]   Junqi Cheng [3]
Yuang Zhang [3]   Zenghui Lu [3]   Huan Yu [3]   Jie Jiang [3]   Peng Shu [3]   Zuxuan Wu [1 2 †]   Yu-Gang Jiang [1 2 †]

## Abstract

Precise camera pose control is critical for video diffusion, yet maintaining geometric consistency remains a challenge. Existing methods that directly inject numerical camera parameters into the diffusion backbone often fail to bridge the gap between abstract coordinates and visual content, leading to structural distortions. To address this issue, we propose CAMERANOISE, a flow-to-noise warping method that encodes camera motion into a temporally coherent stochastic representation. Unlike conventional conditioning, CameraNoise embeds camera poses directly into the noise space. This decouples motion from scene appearance while faithfully preserving trajectory dynamics. Specifically, we introduce a novel Geometry-guided Reprojection Flow and a noise warping algorithm, which jointly preserve the Gaussian prior of diffusion and ensure consistent noise propagation under camera transformations. By integrating CameraNoise into the diffusion process, our framework delivers stable, high-fidelity videos. Extensive experiments demonstrate that our approach significantly outperforms prior methods in both visual quality and trajectory faithfulness. The project page and code are available at: https://gulucaptain.github.io/CameraNoise/.

## 1. Introduction

Camera-controllable video generation has emerged as a core research direction, driven by recent advances in general video diffusion models. Unlike conventional video synthesis, this task aims to generate videos that follow a specific camera trajectory through precise control of camera poses, *i.e.,* the intrinsic and the extrinsic matrices (He et al., 2025b; Zheng et al., 2025). Such control is crucial for applications from personalized video creation and virtual environments to filmmaking, where fine-grained manipulation of viewpoint and motion ensures realism and flexibility.

Despite its broad potential in real-world applications, camera-controllable video generation remains challenging. Achieving realistic results requires temporal smoothness across frames, consistent scene geometry under diverse camera motions, and generalization to arbitrary trajectories. To this end, prior methods typically encode camera poses as Plücker embeddings (He et al., 2025a; Bahmani et al., 2025) or linear feature vectors (Wang et al., 2024b), and inject these numerical representations into the diffusion backbone. While effective, this strategy suffers from two fundamental drawbacks: **1**) they often provide an imprecise representation of camera pose (as shown in cases (a) and (b) in Fig. 1), particularly when capturing subtle lens and variations in speed; **2**) they tend to produce structural distortions and texture discontinuities under new viewpoints.

These issues indicate that feature injection alone is insufficient for reliable camera control. Motivated by recent advances in noise warping (Daras et al., 2024; Chang et al., 2024; Burgert et al., 2025) that leverage optical flow to construct temporally correlated noise fields for motion generation, we propose embedding viewpoint information directly into the noise space as a more principled solution. This ensures that camera conditioning persists throughout the entire generative process rather than being limited to intermediate activations. However, most existing methods (Burgert et al., 2025; Chang et al., 2024), which rely on optical flow derived from object motion, inevitably encode contours and appearance details of the original scene. Consequently, motion information becomes entangled with appearance priors, which can conflict with textual conditions during inference and lead to generation failures (cases (c) and (d) in Fig. 1).

In this paper, we propose a novel camera control framework that introduces a warped CameraNoise to enable precise con-

---

♠Work done during the Tencent Qingyun Program internship. ◇Project lead. †Corresponding authors. [1]Institute of Trustworthy Embodied AI, University of Fudan, Shanghai, China [2]Shanghai Key Laboratory of Multimodal Embodied AI, Shanghai, China [3]Tencent [4]College of Computer Science and Technology, Xiamen University, Fujian, China. Correspondence to: Zuxuan Wu <zxwu@fudan.edu.cn>, Yu-Gang Jiang <ygj@fudan.edu.cn>.

*Proceedings of the $43^{rd}$ International Conference on Machine Learning*, Seoul, South Korea. PMLR 306, 2026. Copyright 2026 by the author(s).

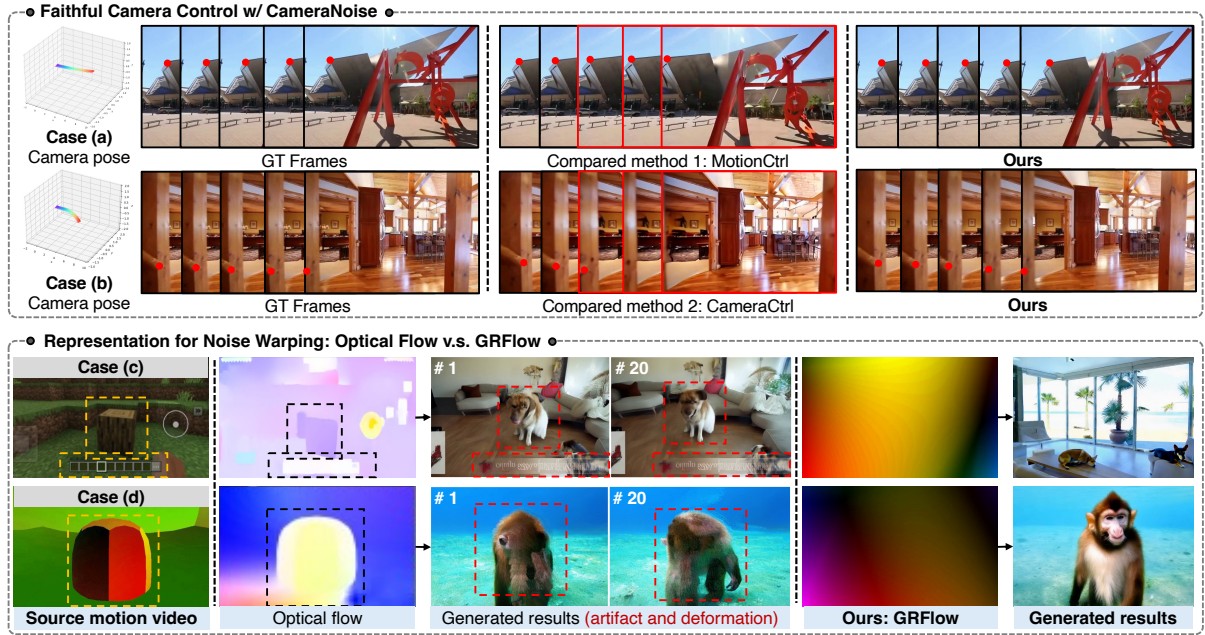

*Figure 1.* We highlight two key observations. (**1**): Shown in case case (a) and (b), camera trajectories visualized by stacked frames with *red anchor points* show that MotionCtrl (Wang et al., 2024b) and CameraCtrl (He et al., 2025a) provide only limited precision, while our method achieves accurate control. (**2**): Comparing optical flow and our GRFlow for CameraNoise warping. Optical flow entangles appearance with motion, causing artifacts (red dashed boxes). By contrast, GRFlow enables appearance-agnostic generation, yielding visually correct and coherent video content. Camera motion templates are obtained from Minecraft (case (c)) and Unreal Engine (case (d)). The textual prompts for cases (c) and (d) are "*Two dogs in a room*" and "*A monkey in the water*", respectively.

trol and high-quality video generation directly in the noise space. Inspired by prior noise warping methods (Chang et al., 2024; Burgert et al., 2025), our CameraNoise retains the advantages of Gaussian noise representations for capturing temporal relations of camera poses, but crucially, it is constructed to be independent of appearance information. This makes CameraNoise inherently appearance-agnostic, allowing it to be fully decoupled from object textures and motion. By injecting CameraNoise into the diffusion process, we achieve camera-controllable video generation while avoiding the semantic conflicts commonly observed in optical-flow-based approaches.

To construct CameraNoise, we introduce Geometry-guided Reprojection Flow (GRFlow), a flow representation that disentangles camera motion from visual content. Unlike optical flow, GRFlow relies solely on camera parameters to characterize pixel displacements across frames. Concretely, we project per-frame camera parameters onto a grid plane and employ Lie algebra optimization to reduce jitter caused by camera pose estimation errors. Building upon GRFlow, we formulate noise warping as a partial differential equation problem and solve it via a bipartite graph. This formulation establishes a one-to-one mapping among camera poses, GRFlow, and CameraNoise. The resulting CameraNoise is then combined with standard Gaussian noise and injected into the diffusion process for camera control. To further

enhance inference robustness, we introduce dynamic perturbations to camera extrinsics during training. We evaluate our method on the RealEstate10K (Zhou et al., 2018), Multi-CamVideo (Bai et al., 2025), and DrivingDoJo (Wang et al., 2024a) datasets across both image- and text-to-video tasks. Experimental results show that our approach achieves more accurate camera control, higher video quality, and stronger visual performance than prior methods, effectively addressing the challenge of imprecise camera control. In short, our work makes the following four contributions:

- We propose a novel camera-controllable diffusion framework that learns camera motion directly from temporal noise. We term this representation Camera-Noise, a warped Gaussian signal that preserves temporal correlations derived from camera parameters.

- We propose an appearance-agnostic Geometry-guided Reprojection Flow (GRFlow) and a corresponding reprojection algorithm for constructing CameraNoise, which disentangles camera motion from scene appearance and prevents semantic conflicts during synthesis.

- We propose formulating CameraNoise warping from GRFlow as a partial differential equation problem, achieving a one-to-one mapping from camera poses to CameraNoise, which ensures precise controllability of the camera motion.

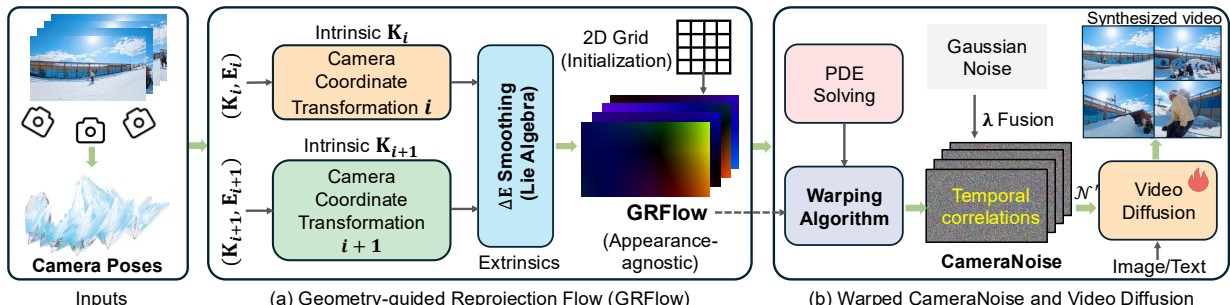

*Figure 2.* Overview of our framework. We introduce CameraNoise, a controlled noise signal that encodes temporal correlations of camera poses into video diffusion. Our method is constructed via Geometry-guided Reprojection Flow (GRFlow) and a Gaussian-preserving warping algorithm, and injected into the video diffusion to enable precise viewpoint control. We use bold green arrows to illustrate the flow of control signals from camera poses to the synthesized video.

- We conduct extensive evaluation to demonstrate the effectiveness of our method, highlighting its ability to generate high-quality videos with faithful and robust camera control.

## 2. Related Work

**Video diffusion.** Since the success and the popularity of the Diffusion Model (Ho et al., 2020), many works pay a lot of effort into achieving video generation (Gu et al., 2023; Chen et al., 2024; Wang et al., 2023b; Zhao et al., 2025; Ni et al., 2024) with deep learning models. In the early stage, video diffusion relies on the pre-trained Stable Diffusion model for image synthesis (Guo et al., 2023; Wang et al., 2023a; Chen et al., 2023). They direct the temporal layer to the SD model to learn the temporal information. Due to the limited performance of temporal consistency, this structure is replaced by video diffusion with 3D convolution and attention layers. Recently, with the growing popularity of DiT model (Peebles & Xie, 2023), several DiT-based diffusion approaches (Yang et al., 2024; Xu et al., 2024; Kong et al., 2024; Wan et al., 2025; Gao et al., 2025) have become mainstream in video generation.

**Camera controllable video generation.** Camera-controllable diffusion model (Liao et al., 2025; Wang et al., 2024b; He et al., 2025a; Zhou et al., 2025) aims to ensure that generated videos follow the motion trajectories defined by camera parameters. To achieve such control, recent method (Wang et al., 2024b) typically encode numerical camera features from intrinsic and extrinsic matrices. In addition to using camera parameter matrices, methods such as Gen3C (Ren et al., 2025) and AC3D (Bahmani et al., 2025) leverage 3D information to improve scene consistency and coherence. A distinct challenge addressed by models like ReCapture (Zhang et al., 2025) and ReCamMaster (Bai et al., 2025) is the re-rendering of existing videos from new camera perspectives. Existing methods inject numerical camera or 3D features into diffusion models to

learn parameter–video mappings, which often leads to weak camera sensitivity and poor discrimination between fast and slow motion. Recent studies (Chang et al., 2024; Burgert et al., 2025) show that injecting temporal information into the latent can steer video generation toward specific motion patterns. Inspired by this, we propose a novel approach that directly controls camera movement in the noise space.

**Camera pose estimation.** When generating videos from camera parameters, control conditions typically rely on camera estimation algorithms (Zhao et al., 2026; Camposeco et al., 2018; Schonberger & Frahm, 2016; Li et al., 2026) because most online videos including camera control templates lack explicit camera parameter annotations. Even the RealEstate10K dataset (Zhou et al., 2018) uses model-based estimation to generate its annotations. COLMAP (Schonberger & Frahm, 2016) is the most widely used traditional pipeline, comprising multiple stages such as image matching, triangulation, and bundle adjustment. Among recent methods, VGGT (Wang et al., 2025) achieves state-of-the-art results by directly estimating camera parameters through extensive parameter fitting. Given these advantages, we adopt VGGT for camera pose estimation and evaluation.

## 3. Method

Our primary objective is to enable camera-controllable video generation while mitigating imprecise control from numerical embedding injections and distortions during generation. As shown in Fig. 2, our framework encodes camera motion patterns directly in the noise space via GRFlow and CameraNoise, enabling precise viewpoint control without interfering with the denoising process.

### 3.1. Geometry-guided Reprojection Flow

Since CameraNoise models pixel motion across frames, we first introduce an appearance-agnostic Geometry-guided Reprojection Flow (GRFlow), denoted as $\mathcal{G}_r$, along with its

corresponding reprojection algorithm. GRFlow defines a precise mapping from camera poses to geometric transformations, effectively decoupling motion information from appearance cues, a problem that typically arises in optical-flow-based methods. Specifically, we take camera parameters $[\mathbf{K}_i, \mathbf{E}_i]$ for the source frame and $[\mathbf{K}_{i+1}, \mathbf{E}_{i+1}]$ for the target frame, where $\mathbf{K} \in \mathbb{R}^{3 \times 3}$ denotes the intrinsic matrix and $\mathbf{E} = [\mathbf{R}; \mathbf{t}] \in \mathbb{R}^{3 \times 4}$ the extrinsic matrix. Together, the pair $(\mathbf{R}, \mathbf{t})$ fully determines the camera's imaging process in the world coordinate system $\mathbf{R}^w$.

**Reprojection using camera poses.** Let the 2D image plane be a discrete sampling of a continuous manifold $\mathcal{I} \subset \mathbb{R}^2$, represented by the grid $\Omega \subset \mathcal{I}$. The camera extrinsic pose $\mathbf{E} = [\mathbf{R}; \mathbf{t}]$ is represented as an element of SE(3), the Special Euclidean group, which acts on the 3D scene manifold $\mathcal{S} \subset \mathbb{R}^3$. We define the GRFlow as:

$$\mathcal{G}_r : \Omega \times \mathrm{SE}(3) \to \Omega, (x, y; \mathbf{E}_i, \mathbf{E}_{i+1}) \mapsto (x', y'), \quad (1)$$

that encodes the effect of rigid camera motion on 2D pixel positions without relying on scene appearance (Proposition 1). We denote $(x, y)$ as a pixel from the source frame, which corresponds to the pixel $(x', y')$ on the target frame. Each pixel at $(x, y)$ is represented in homogeneous form as $\Omega_{x,y} = [x, y, 1]^\top \in \mathbb{P}^2$, where $\mathbb{P}^2$ means the 2D projective space. These pixels define rays in the camera coordinate via the inverse intrinsic matrix. To reconstruct the pseudo-3D points, we rely solely on camera poses and back-project each 2D pixel to the camera coordinate frame using a constant pseudo-depth $d$. We adopt a homogeneous column-vector convention and denote the homogeneous image coordinate as $\mathbf{\Omega}_{x,y} = [x, y, 1]^\top$. The lifted 3D point is then computed:

$$\mathbf{p}_{x,y} = d\hat{\mathbf{x}}_{x,y}, \quad \hat{\mathbf{x}}_{x,y} = \mathbf{K}_i^{-1}\mathbf{\Omega}_{x,y} \in \mathbb{R}^3. \quad (2)$$

We formalize Eq. (2) as a lifting map that maps a 2D pixel to a pseudo-3D point along its viewing ray:

$$\ell : \Omega \to \mathbb{R}^3, \quad \ell(x, y) = d\mathbf{K}_i^{-1}[x, y, 1]^\top. \quad (3)$$

Then, we extend the 3D point to homogeneous coordinates as $\tilde{\mathbf{p}}_{x,y} = [\mathbf{p}_{x,y}^\top, 1]^\top \in \mathbb{R}^4$. Given the homogeneous transformation $\mathbf{E}_i \in \mathrm{SE}(3)$, the relative transformation from the source frame $i$ to the target frame $i + 1$ under the column-vector convention is defined as:

$$\Delta\mathbf{E}_{i \to i+1} = \mathbf{E}_{i+1}\mathbf{E}_i^{-1}. \quad (4)$$

The lifted point is then mapped to the target camera coordinate frame by:

$$\tilde{\mathbf{p}}'_{x,y} = \Delta\mathbf{E}_{i \to i+1}\tilde{\mathbf{p}}_{x,y}. \quad (5)$$

**$\Delta\mathbf{E}$ smoothing via Lie algebra.** In practice, errors in camera pose estimation are inevitable, leading to discontinuities

> **Proposition 1** (Appearance Agnostic). *GRFlow is computed only from camera poses, so it captures geometric motion while remaining independent of visual appearance.*
>
> **Proposition 2** (Approximation of Continuous Flow). *Under smoothed camera motion and typical latent diffusion resolution, GRFlow converges pointwise to the continuous flow map, with discretization bounding the error.*

in $\Delta\mathbf{E}$ and causing irregular jitter in GRFlow. Since the rotation matrix $\mathbf{R} \in \mathrm{SO}(3)$ is subject to nonlinear constraints, direct optimization in the matrix space makes it difficult to preserve the group structure. To address this issue, we propose a smoothing function for $\Delta\mathbf{E}$ based on Lie algebra, which maps nonlinear transformation matrices into a linear space for processing. Specifically, we extract the rotational vector $\omega_i \in \mathrm{so}(3)$ from $\mathbf{R}_i$ and compute the rotation angle:

$$\cos\theta_i = (\mathrm{tr}(\mathbf{R}_i) - 1)/2 \implies \theta_i = \arccos(\cos\theta_i). \quad (6)$$

We calculate the unit vector of the rotation axis $\mathbf{k}_i$ and get the rotational vector as: $\omega_i = \theta_i \times \mathbf{k}_i$. The extrinsic matrix $\mathbf{E}_i$ is then represented with Lie Algebra vector as $\xi_i = [\boldsymbol{\omega}_i, \boldsymbol{t}_i] = [\omega_{i,x}, \omega_{i,y}, \omega_{i,z}, t_{i,x}, t_{i,y}, t_{i,z}] \in \mathrm{se}(3)$, where $\{\omega_{i,x}, \omega_{i,y}, \omega_{i,z}\}$ are rotate vectors and $\{t_{i,x}, t_{i,y}, t_{i,z}\}$ are translations. For small rotations ($\theta \approx 0$), we set $\omega_i = 0$. This Lie algebra representation enables smoothing of the trajectory using an exponentially weighted moving average with factor $\alpha$:

$$\xi_i^{\mathrm{smooth}} = \begin{cases} \xi_i, & i = 1 \\ \alpha \cdot \xi_i + (1 - \alpha) \cdot \xi_{i-1}^{\mathrm{smooth}}, & i > 1. \end{cases} \quad (7)$$

To incorporate the smoothed parameters into the GRFlow, we apply the exponential map to convert the $\xi_i^{\mathrm{smooth}}$ back into a transformation matrix $\Delta\mathbf{E}$ in the SE(3) space.

**GRFlow construction.** The transformed 3D point is computed by matrix multiplication in homogeneous coordinates: $\tilde{\mathbf{p}}'_{x,y} = \Delta\mathbf{E} \times \tilde{\mathbf{p}}_{x,y}$. This operation is equivariant under SE(3), meaning that the transformation preserves rigid-body geometry:

$$\|\mathbf{p}'_1 - \mathbf{p}'_2\|_2 = \|\mathbf{p}_1 - \mathbf{p}_2\|_2, \quad \forall \mathbf{p}_1, \mathbf{p}_2 \in \mathbb{R}^3. \quad (8)$$

Subsequently, we reproject points onto the 2D image plane of the target camera: $\Omega'_{x,y} = \mathbf{K}_{i+1} \times \tilde{\mathbf{p}}'_{x,y} \in \mathbb{P}^2$, and obtain the Cartesian coordinates $(x', y')$ by normalizing homogeneous coordinates. The particle flow on the source image, $\mathcal{F}_{x,y}$, is then defined as the displacement between the target and source pixel locations: $\mathcal{F}_{x,y} = (x', y') - (x, y)$. Iterating this flow across all frames generates the full GRFlow, which pushes forward the pixel coordinates through the rigid-body action:

$$\mathcal{G}_r = \{\mathcal{F}_{x,y}^{i \to i+1} \mid (x, y) \in \Omega, i = 1, \ldots, T - 1\}. \quad (9)$$

Noticed the GRFlow is not intended to reconstruct exact 3D scene dynamics; instead, it serves as a camera-motion-specific geometric prior for noise warping. From this perspective, Eq. 2 with a pseudo-depth $d$ is designed to capture the dominant global effect of relative camera motion while remaining scene-agnostic and free from appearance or object-motion leakage. Furthermore, we show that GRFlow serves as an approximation of continuous flow (Proposition 2), and we provide a formal proof in the Appendix B.

### 3.2. The CameraNoise Representation

The main purpose of CameraNoise is to encode temporal priors of camera motion into Gaussian noise using GR-Flow. Specifically, GRFlow propagates the initial noise such that it remains temporally consistent while preserving its Gaussian distribution. However, a naive interpolation between adjacent frame pixels would violate this prior (see Appendix C.1). To address the issue, we introduce a novel warping algorithm to generate CameraNoise, which formulates noise propagation as the discrete solution of advection *Partial Differential Equations (PDEs)*. Furthermore, we incorporate a density correction mechanism to uniformly accommodate diverse camera motion patterns.

Given an initial noise frame $q_t \in \mathbb{R}^{H \times W}$ and the advection vectors for each grid point in GRFlow $\mathcal{G}_r$, we aim to generate the subsequent noise frame $q_{t+1}$. The propagation from $q_t$ to $q_{t+1}$ is constrained by the following principles: **1)** Temporal continuity: the next-frame noise depends solely on the state of the previous frame; **2)** Preservation of the Gaussian prior: the noise distribution remains a standard Gaussian. We formulate noise propagation as a linear advection PDE:

$$\frac{\partial \rho(x,t)}{\partial t} + v(x,t) \cdot \nabla \rho(x,t) = 0, \quad v(x,t) = \mathcal{G}_r^t, \quad (10)$$

where $\rho(x,t)$ denotes the probability density of the noise field at position $x$ and time $t$, and $v(x,t)$ represents the velocity field given by $\mathcal{G}_r$. This equation describes the advection of the noise density $\rho$ by the velocity field $v$, where volume changes modulate the local noise concentration. To solve the PDE on discrete frames, we adopt a bipartite graph formulation combined with a density-weighted discretization strategy. In this graph, nodes on the left correspond to positions $\rho_t(y)$ in the current noise frame, while nodes on the right represent positions $\rho_{t+1}(x)$ in the subsequent frame. Edges $(y \to x)$ encode the transport paths determined by GRFlow. On a discrete grid, the continuous flow is represented as correspondences between pixels, which can be categorized into two motion types: expansion and contraction. For the edge weights $w(x,y)$, we compute either the local flow density or the determinant of the Jacobian matrix $\det J(x)$ (details in Appendix D), which quantifies the local degree of expansion and contraction in the noise

field. For the backward flow in GRFlow, we directly use $-\mathcal{G}_r$. The divergence term $\nabla \cdot (\rho v)$ in the PDE is discretized as a mass-transfer operation over the graph edges. Then, density normalization is computed as:

$$\rho_{t+1}(x) = \sum_{y \mapsto x} w(x,y)\rho_t(y) \, / \sum_{y \mapsto x} w(x,y). \quad (11)$$

By leveraging a bipartite graph and sparse matrix-based transport, we reformulate the PDE as a linear algebra problem defined on the graph. This warping algorithm circumvents the complexity of directly solving the PDE, yielding a stable and computationally efficient discrete solution. By iterating this process across frames, we generate a temporally consistent noise sequence, which we denote as CameraNoise. We further demonstrate that the mapping from camera poses to CameraNoise is one-to-one (Proposition 3), with the corresponding proof provided in Appendix C.2.

> **Proposition 3** (One-to-One Mapping). *By the PDE uniqueness theorem, each $\rho$ has a unique solution $\rho(x)$, making the mapping from GRFlow to CameraNoise bijective and ensuring a one-to-one correspondence between camera poses and CameraNoise.*

### 3.3. Video Diffusion Model with CameraNoise

In this subsection, we introduce CameraNoise into the diffusion model to achieve camera-controllable video generation. After undergoing GRFlow reprojection and PDE solving, CameraNoise encodes temporally coherent information in the noise space that is equivalent to camera motion. This enables it to guide the motion direction of the generated frames during inference, thereby simulating camera movements. To integrate CameraNoise $\mathcal{N}_C$ into the diffusion process, we fuse it with random Gaussian noise $\mathcal{N}$ during training with:

$$\mathcal{N}' = (\mathcal{N} \cdot \lambda + \mathcal{N}_C \cdot (1-\lambda))/\sqrt{\lambda^2 + (1-\lambda)^2}, \quad (12)$$

where $\lambda$ is a mixing coefficient that regulates the relative contribution of the two components. During training, we randomly sample $\lambda$ from the interval $[0,1]$ for each training case. This strategy ensures that the model remains sensitive to both the standard Gaussian distribution and CameraNoise. Because CameraNoise encodes camera-specific information, directly replacing the standard Gaussian with CameraNoise in the diffusion model would compromise the model's prior knowledge. During inference, the fusion coefficient is set to zero or a small value, ensuring stable control.

Moreover, in the training phase, we introduce a Dynamic Scaling Training (DST) strategy for extrinsic matrices to enhance the robustness of camera parameter estimation. For rotate matrix $\mathbf{R}$ in the extrinsic, we denote the $\mathbf{R} =$

*Table 1.* Quantitative comparisons with MotionCtrl (Wang et al., 2024b), CameraCtrl (He et al., 2025a), and AC3D (Bahmani et al., 2025) methods on RealEstate100 test set, including Text-to-Video (T2V) and Image-to-Video (I2V) generation.

| Methods | Aesthetic Quality ↑ | Imaging Quality ↑ | Motion Smoothness | Dynamic Degree | TransErr ↓ | RotErr ↓ | LPIPS ↓ | FVD ↓ |
|---|---|---|---|---|---|---|---|---|
| MotionCtrl-T2V | 0.521 | 0.731 | 0.980 | 0.10 | 0.372 | 0.476 | **0.468** | 661.04 |
| CameraCtrl-T2V | 0.403 | 0.706 | 0.974 | 0.26 | 0.344 | 0.483 | 0.556 | 909.58 |
| AC3D | 0.476 | 0.613 | **0.993** | 0.15 | 0.391 | 0.472 | 0.634 | 800.79 |
| **Ours-T2V** | **0.549** | **0.740** | **0.993** | **0.53** | **0.232** | **0.436** | 0.528 | **437.54** |
| MotionCtrl-I2V | 0.414 | 0.654 | 0.992 | 0.33 | 0.445 | 0.606 | 0.312 | 321.79 |
| CameraCtrl-I2V | 0.482 | 0.672 | **0.993** | 0.17 | 0.173 | 0.276 | 0.258 | 392.79 |
| **Ours-I2V** | **0.509** | **0.689** | **0.993** | **0.58** | **0.144** | **0.269** | **0.182** | **180.81** |

$\exp(\theta[\mathbf{u}]_\times)$, with Rodrigues function (Dai, 2015), where $\mathbf{u} \in \mathbb{R}^3$, $\|\mathbf{u}\|$ denotes the rotary axis, and $\theta \in \mathbb{R}$ represents the rotation angle. The $[\mathbf{u}]_\times \in \mathbb{R}^{3\times 3}$ is the antisymmetric matrix of $\mathbf{u}$. We denote the rotational vector $\mathbf{r}$ as: $\mathbf{r} = \theta\mathbf{u}$ and the rotate matrix is:

$$\mathbf{R} = \exp(\theta[\mathbf{u}]_\times) = \exp([\mathbf{r}]_\times), \qquad (13)$$

with $\mathbf{r} = (r_x, r_y, r_z)^\top, \theta = \sqrt{r_x^2 + r_y^2 + r_z^2}$. We define the scale factor $\eta$ for $\mathbf{r}$ as $\mathbf{r}' = \epsilon \times \mathbf{r} = (\eta \times \theta) \times \mathbf{u}$. Then, we can obtain rescaled $\mathbf{R}'$:

$$\mathbf{R}' = \exp([\mathbf{r}']_\times) = \exp((\eta \cdot \theta) \cdot [\mathbf{u}]_\times). \qquad (14)$$

We maintain the translation part $\mathbf{t}$ and update the extrinsic parameter as: $\mathbf{E} = [\mathbf{R}'; \mathbf{t}]$. In practice, we control the factor $\eta$ manually, and we set $\eta \in (0.9, 1.1)$ for diffusion training. This scaling strategy improves model performance during inference with estimated camera poses, particularly enhancing robustness across varying rotation angles.

## 4. Experiments

### 4.1. Implementation Details

**Dataset.** Our model is trained on the combined data from the RealEstate10K dataset (Zhou et al., 2018) and the DynPose-100K dataset (Rockwell et al., 2025). The RealEstate10K contains approximately 10 million frames with associated camera poses. The DynPose-100K is a large-scale collection of dynamic internet videos annotated with camera poses, featuring a diverse range of attributes. Following prior works (He et al., 2025a; Wang et al., 2024b), we evaluate on 100 randomly selected samples from the RealEstate10K, denoted as *RealEstate100*, with sample IDs provided in Appendix I. For a fair comparison, the model evaluated on RealEstate100 is trained only on RealEstate10K. Furthermore, we evaluate the performance on two dynamic and outdoor datasets, MultiCamVideo (Bai et al., 2025) and DrivingDoJo (Wang et al., 2024a). We also take 100 random examples from each dataset as the test sets, termed *MultiCamVideo100* and *DrivingDoJo100*.

**Evaluation metrics.** We evaluate our model and baselines using two groups of metrics. For frame-level quality, we report *LPIPS* (Zhang et al., 2018). For video-level quality, we use *Fréchet Video Distance (FVD)* (Unterthiner et al., 2018) and VBench (Huang et al., 2024), the latter offering targeted evaluation with custom prompts and metrics. We focus on dimensions most relevant to video generation, including *Aesthetic Quality, Imaging Quality, Motion Smoothness, and Dynamic Degree*. To further assess camera motion, we employ the *TransErr* and *RotErr* (He et al., 2025a) metrics. Due to page limitations, we demonstrate more implementation details in Appendix E.

**Baselines.** We mainly compare against four camera-control baselines: MotionCtrl (Wang et al., 2024b), CameraCtrl (He et al., 2025a), and AC3D (Bahmani et al., 2025). Noticed that MotionCtrl, CameraCtrl, and our method support both image-to-video and text-to-video generation. Moreover, we also evaluate the warped-noise-based Go-with-the-Flow (GWTF) (Burgert et al., 2025) method and 3D-based GEN3C (Ren et al., 2025) method.

### 4.2. Comparisons with Other Methods

**Quantitative comparison.** Table 1 presents model performance on Text-to-Video (T2V) and Image-to-Video (I2V) tasks using the RealEstate100 test set, comparing our method with four baselines. "Model"-T2V refers to inference conditioned solely on text, while "Model"-I2V refers to inference conditioned on an image or both image and text. Our model achieves the best overall performance in both video quality and camera control. In terms of video quality, it demonstrates strong results in image quality, temporal consistency, and dynamism. Notably, for I2V, our method increases the dynamic degree by 75.8% over the previous best approach, while maintaining motion smoothness. Our model also achieves state-of-the-art performance in camera motion evaluation compared to existing methods.

Furthermore, Table 2 reports the zero-shot I2V results on the MultiCamVideo100 and DrivingDojo100 test sets. We mark the optimal results in bold and the sub-optimal results

*Table 2.* Zero-shot I2V comparisons with MotionCtrl, CameraCtrl, Go-with-the-Flow (GWTF) (Burgert et al., 2025), and GEN3C (Ren et al., 2025) methods on the MultiCamVideo100 (upper block) and DrivingDojo100 (lower block) test sets.

| Methods | Aesthetic Quality ↑ | Imaging Quality ↑ | Motion Smoothness | Dynamic Degree | TransErr ↓ | RotErr ↓ | LPIPS ↓ | FVD ↓ |
|---|---|---|---|---|---|---|---|---|
| MotionCtrl-I2V | 0.535 | **0.707** | 0.991 | 0.02 | 0.287 | 0.556 | 0.558 | 908.01 |
| CameraCtrl-I2V | 0.571 | 0.537 | 0.992 | 0.02 | 0.242 | 0.30 | 0.561 | 929.44 |
| GEN3C | 0.546 | 0.571 | 0.988 | 0.13 | 0.212 | 0.343 | 0.521 | 724.74 |
| GWTF | 0.592 | 0.594 | 0.987 | **0.26** | **0.187** | 0.235 | 0.566 | 667.68 |
| **Ours** | **0.601** | 0.623 | **0.993** | 0.25 | 0.191 | **0.230** | **0.298** | **362.57** |
| MotionCtrl-I2V | 0.428 | **0.575** | 0.991 | 0.22 | 1.445 | 0.331 | 0.505 | 672.90 |
| CameraCtrl-I2V | 0.448 | 0.559 | **0.994** | 0.04 | 2.140 | **0.210** | 0.493 | 667.64 |
| GEN3C | 0.440 | 0.521 | 0.985 | 0.45 | 0.937 | 0.320 | 0.482 | 374.83 |
| GWTF | 0.458 | 0.499 | 0.986 | **0.88** | 0.776 | 0.359 | 0.496 | 341.29 |
| **Ours** | **0.461** | 0.536 | 0.984 | **0.88** | **0.397** | 0.265 | **0.158** | **242.67** |

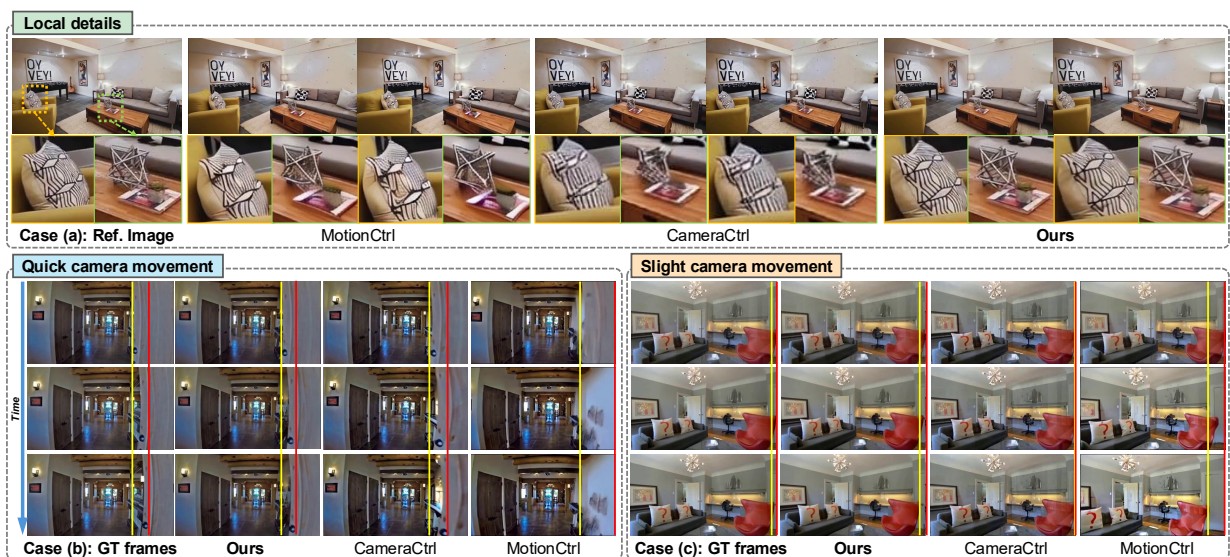

*Figure 3.* Qualitative comparison on RealEstate10K with MotionCtrl (Wang et al., 2024b) and CameraCtrl (He et al., 2025a). We illustrate two aspects: **1**) visual quality and local details for case (a); **2**) camera movement under fast and slight motion in cases (b) and (c), where vertical **red** and **yellow** lines mark temporal anchor points. Their gap reflects motion magnitude relative to the ground truth.

with an underline. Since these models are trained on different datasets, the zero-shot experiments provide a more objective assessment of the models' generalization ability. The results show that our method also achieves the best or highly competitive performance on these dynamic-scene datasets. This improvement results from the proposed CameraNoise, which directly controls camera motion at the noise level. Combined with our camera pose warping algorithm, it effectively mitigates noise and non-expressive artifacts.

**Qualitative comparison.** Fig. 3 presents a visual comparison with prior approaches using data sampled from RealEstate10K. Our model significantly outperforms existing methods in preserving fine details within frames and accurately representing camera motion, including rapid and slight camera movement. In addition, **we provide extensive qualitative visualizations and corresponding analyses in Appendix A**, including dynamic scenarios (in Fig. 7), out-

of-distribution scenarios (in Fig. 8), long-video generation (in Fig. 9), diverse camera motion (in Fig. 10), and cross-scene camera motion transfer (in Fig. 11), to further comprehensively evaluate the generalization ability and stability of our model under diverse and challenging conditions.

**Comparison between GRFlow and Optical flow.** In Fig. 4, we compare the visualization results of the proposed GRFlow with those of conventional optical flow in dynamic scenes. It can be observed that GRFlow is not designed to capture object-level appearance motion in the scene, but rather to explicitly characterize the variations of the camera itself. Accordingly, we deliberately decouple GRFlow from scene content during modeling, so that it reflects only the global geometric transformations induced by camera motion. This design effectively prevents the introduction of unintended appearance priors during the mapping from GRFlow to CameraNoise, thereby avoiding the erroneous in-

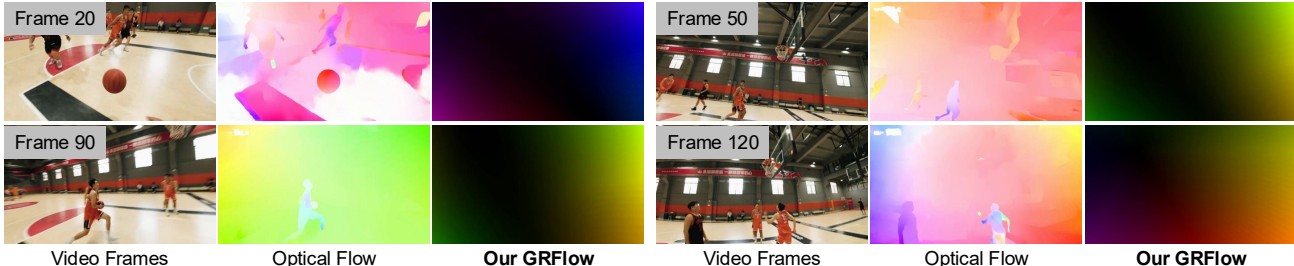

Frame 20    Frame 50    Frame 90    Frame 120

Video Frames    Optical Flow    **Our GRFlow**    Video Frames    Optical Flow    **Our GRFlow**

*Figure 4.* Qualitative comparison between optical flow (Teed & Deng, 2020) and our proposed GRFlow in a real dynamic scenario. GRFlow is designed to capture camera motion information from video frames, while optical flow struggles to separate camera motion from scene appearance. This design ensures that GRFlow does not introduce appearance priors during CameraNoise warping.

*Table 3.* Effectiveness of the quality of GRFlow with the smoothing algorithm for video generation. We use a setting with $\alpha = 1.0$ as the baseline (in gray), which corresponds to results without smoothing.

| Methods | Aesthetic Quality ↑ | Imaging Quality ↑ | Motion Smoothness | Dynamic Degree | TransErr ↓ | RotErr ↓ | FVD ↓ |
|---|---|---|---|---|---|---|---|
| $\alpha = 1.0$ | 0.517 | 0.707 | 0.994 | 0.51 | 0.145 | 0.281 | 181.63 |
| $\alpha = 0.8$ | -0.19% | +0.52% | +0.0% | +1.96% | +0.54% | -8.71% | +7.86% |
| $\alpha = 0.6$ | -0.19% | +0.29% | +0.0% | -3.92% | +3.92% | +2.28% | +1.17% |
| $\alpha = 0.4$ | -0.19% | -0.14% | +0.0% | -7.84% | +0.69% | +4.15% | +0.77% |
| $\alpha = 0.2$ | -15.47% | -2.57% | -0.10% | +13.73% | +0.69% | +4.15% | +5.66% |
| $\alpha = 0.0$ | -13.54% | +1.84% | +0.10% | -37.25% | -376.19% | -211.72% | -54.61% |

jection of scene content into the noise space and preserving the stability and generation quality of the video diffusion.

### 4.3. Ablation Study

**Impact of GRFlow quality on video results.** In Table 3, we show how different $\alpha$ values in the $\Delta\mathbf{E}$ smoothing algorithm affect the quality of GRFlow and, consequently, the video generation results. Notably, all quantitative ablation studies are on the RealEstate100 test set for I2V task. The results show that decreasing $\alpha$ (*i.e.,* strengthening the smoothing effect) leads to a notable improvement in generation quality. However, while $\alpha$ can be set arbitrarily close to 0, it cannot be exactly 0, as this would disrupt the numerical structure of $\Delta\mathbf{E}$ and cause a sharp degradation in performance.

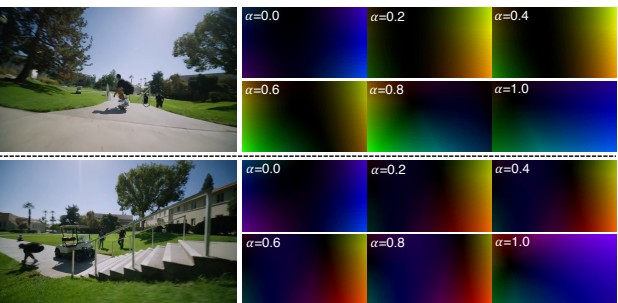

*Figure 5.* Ablation of GRFlow with different $\alpha$ values.

**Effect of $\alpha$ value in $\Delta\mathbf{E}$ smoothing on GRFlow.** During the generation of GRFlow, the smoothing algorithm's $\alpha$ pa-

rameter plays a critical role in shaping the final results. As shown in Fig. 5, varying $\alpha$ lead to distinct outcomes, *i.e.,* the changing of GRFlow with different values, highlighting its influence under real-world motion scenarios. This demonstrates that careful tuning of $\alpha$ is essential for achieving stable and accurate camera motion representation.

**Impact of the fusion ratio $\lambda$ on video results.** Table 4 reports inference performance under different value $\lambda$ in Eq.12, which controls the mixing ratio between CameraNoise and Gaussian noise. As the ratio increases, we observe slight improvements in aesthetic quality and imaging quality, but substantial drops in video dynamism and camera-control accuracy. From the table, a clear trend can be observed: as the proportion of CameraNoise decreases, the effectiveness of camera control drops significantly. Specifically, when $\lambda$ is 0.6, the camera control performance declines by nearly 50%. This result not only quantifies the impact of noise composition on generation but also strongly validates the effectiveness of our approach for controlling camera trajectories in the noise space.

**Effect of the dynamic scaling strategy during training.** We introduce camera control signals into the diffusion process via CameraNoise and propose a dynamic scaling training (DST) strategy to adaptively adjust the scaling of rotation matrices in camera extrinsics during training. As shown in Table 5, we conduct a control experiment without the DST strategy and systematically evaluate the effect of varying parameter $\eta$ within the DST strategy. The results

*Table 4.* Effectiveness of the weight $\lambda$, which controls the mixing ratio between Gaussian noise and CameraNoise during inference. We use $\lambda = 0$ as the baseline, where CameraNoise alone initializes the noise, and gradually increase $\lambda$ to introduce more Gaussian noise.

| Methods | Aesthetic Quality ↑ | Imaging Quality ↑ | Motion Smoothness | Dynamic Degree | TransErr ↓ | RotErr ↓ | FVD ↓ |
|---|---|---|---|---|---|---|---|
| $\lambda = 0.0$ | 0.509 | 0.689 | 0.993 | 0.58 | 0.153 | 0.248 | 196.83 |
| $\lambda = 0.1$ | +0.79% | +1.60% | +0.10% | -5.17% | -0.85% | -5.96% | +0.57% |
| $\lambda = 0.2$ | +0.79% | +1.89% | +0.0% | -5.17% | +5.62% | -8.53% | +12.95% |
| $\lambda = 0.4$ | +1.57% | +1.89% | +0.10% | -3.44% | -6.70% | -17.48% | +6.95% |
| $\lambda = 0.6$ | +1.38% | +1.74% | +0.0% | -8.62% | -45.15% | -57.04% | -5.67% |
| $\lambda = 0.8$ | +1.57% | +4.21% | +0.0% | -6.90% | -135.52% | -123.38% | -30.55% |
| $\lambda = 1.0$ | +2.36% | +4.79% | +0.0% | -12.07% | -145.30% | -121.16% | -22.04% |

*Table 5.* Impact of the dynamic scaling training (DST) strategy. $\eta^*$ denotes a fixed value.

| Methods | Imaging Quality ↑ | TransErr ↓ | RotErr ↓ | FVD ↓ |
|---|---|---|---|---|
| w/o DST | 0.630 | 0.167 | 0.304 | 198.04 |
| w/ $\eta^* = 0.9$ | 0.675 | 0.155 | 0.277 | 188.36 |
| w/ $\eta \in (0.8, 1.2)$ | 0.642 | 0.160 | 0.270 | 182.95 |
| w/ $\eta \in (0.9, 1.1)$ | **0.689** | **0.144** | **0.269** | **180.81** |

show that the adjustments of DST can effectively enhance the model's robustness during inference.

*Table 6.* Scalability analysis of the proposed graph-based PDE solver under different video resolutions (Reso.). The solver operates in the latent space with an $8\times$ spatial downsampling ratio. FPS denotes the processing speed of the PDE solver.

| Video Reso. | Latent Reso. | FPS ↑ | Time / Frame (s) ↓ | Memory (MB) ↓ |
|---|---|---|---|---|
| $1024 \times 576$ | $72 \times 128$ | 20.17 | 0.050 | 644.95 |
| 1080p | $135 \times 240$ | 13.15 | 0.076 | 764.38 |
| 4K | $270 \times 480$ | 5.60 | 0.179 | 1574.64 |

**Scalability of the Graph-based PDE Solver.** To evaluate the scalability of the proposed graph-based PDE solver, we report its computational cost under different video resolutions. Since modern video diffusion backbones perform denoising in the latent space, the actual resolution processed by our CameraNoise solver is substantially smaller than the original video resolution. In our implementation, the backbone adopts an $8\times$ spatial downsampling ratio, resulting in latent resolutions of $72 \times 128$, $135 \times 240$, and $270 \times 480$ for $1024 \times 576$, 1080p, and 4K videos, respectively. As shown in Table 6, the proposed PDE solver scales efficiently with the latent resolution. Even for 4K video generation, the solver achieves 5.60 FPS, corresponding to only 0.179 seconds per frame, with 1574.64 MB CPU memory usage. Moreover, the solver runs entirely on the CPU and therefore introduces no additional GPU memory overhead beyond the diffusion backbone. These results demonstrate that the proposed graph-based PDE solver for CameraNoise warping is computationally lightweight and practical for high-resolution video generation.

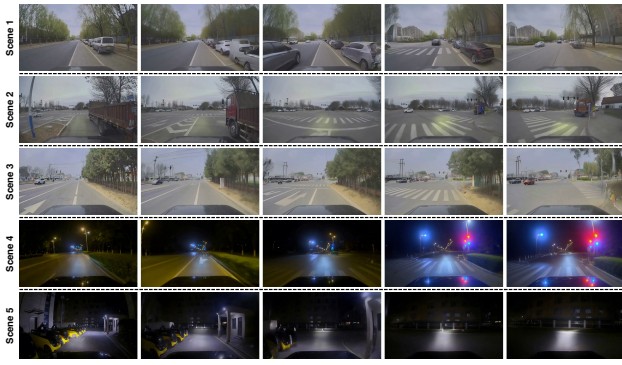

*Figure 6.* Qualitative results of our method with CameraNoise in driving scenarios. Timeline expands horizontally.

### 4.4. More Applications

Beyond general camera-controlled video generation, we further explore the applicability of our model to driving scenarios under the image-to-video setting, as shown in Fig. 6. During testing, we combine driving-scene images from the DrivingDoJo dataset (Wang et al., 2024a) with specified camera motion trajectories to generate videos across five different scenarios. These results demonstrate that our model can generalize camera-controllable generation to complex driving environments, where scene layouts, illumination, and depth structures vary significantly.

## 5. Conclusion

In this work, we introduced CameraNoise, a stochastic representation that embeds camera poses into the noise space independently of scene appearance, enabling temporally coherent video generation. Leveraging a Geometry-guided Reprojection Flow (GRFlow) and a CameraNoise warping algorithm, our method preserved the Gaussian prior of the diffusion process and maintains consistent noise propagation under camera transformations. Integrating CameraNoise into the video diffusion enables precise camera control, producing high-quality and stable videos. Experimental results showed that our approach achieved higher video quality and stronger visual performance than prior methods, effectively addressing the challenge of imprecise camera control.

## Acknowledgements

This work is supported by the National Natural Science Foundation of China (Grant No. 62521004), the Science and Technology Commission of Shanghai Municipality (No. 25511106100), and the New Cornerstone Science Foundation through the XPLORER PRIZE.

## Impact Statement

This paper presents work whose goal is to advance the fields of Machine Learning and Deep Learning. There are many potential societal consequences of our work, none which we feel must be specifically highlighted here.

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

# A. More Main Results.

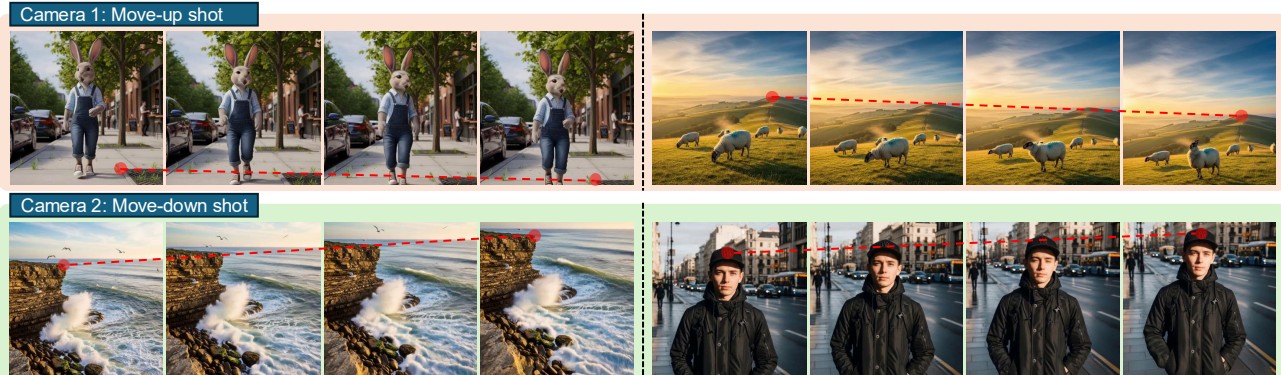

*Figure 7.* Dynamic results across multiple scenes under different camera poses. In each scene, an anchor point is highlighted with a red circle, and dashed lines indicate the camera's motion trajectory.

**Qualitative results under dynamic scenarios.** To validate the performance of our model in dynamic scenarios, where the scene involves not only camera motion but also dynamic object motion, we conduct comparative experiments under two different camera motions across four representative scenarios, as shown in Fig. 7. The results indicate that our model is capable of handling not only camera motion in largely static scenes such as RealEstate10K (Zhou et al., 2018), but also maintaining strong performance in dynamic scenes with substantial object motion.

*Table 7.* Quantitative analysis of CameraCtrl (He et al., 2025a), MotionCtrl (Wang et al., 2024b), Go-with-the-Flow (Burgert et al., 2025), and GEN3C (Ren et al., 2025) in dynamic OOD scenarios with image-to-video generation. Since there are no ground-truth videos available for these OOD scenarios, we evaluate the models using video quality metrics provided by VBench (Huang et al., 2024). The optimal results are in bold, while the sub-optimal results are marked with underlining.

| Methods | Aesthetic Quality ↑ | Imaging Quality ↑ | Motion Smoothness | Dynamic Degree |
|---|---|---|---|---|
| CameraCtrl | 0.635 | 0.578 | 0.991 | 0.0 |
| MotionCtrl | 0.633 | 0.695 | 0.987 | 0.0 |
| Go-with-the-Flow | 0.636 | 0.597 | 0.986 | **0.33** |
| GEN3C | 0.652 | 0.651 | **0.992** | 0.17 |
| **CameraNoise** | **0.712** | **0.707** | 0.991 | 0.25 |

**Performances under out-of-distribution (OOD) scenarios.** Out-of-Distribution (OOD) scenarios are commonly used as important data to evaluate model robustness. To this end, we evaluate the performance of MotionCtrl (Wang et al., 2024b), CameraCtrl (He et al., 2025a), Go-with-the-Flow (Burgert et al., 2025), GEN3C (Ren et al., 2025), and our method across six types of scenes: valleys, fields, lakes, deserts, forests, and amusement parks. For these scenes, we apply six typical camera motions, including move-up, move-down, move-left, move-right, move-clockwise, and zoom-in shots, which are also the typical camera movements specified in the GEN3C method. Table 7 presents the quantitative analysis of these methods across the scenes, and Fig. 8 visualizes four selected scenarios.

From these OOD quantitative and qualitative results, we summarize the characteristics and limitations of the referred mainstream methods under OOD settings: 1) MotionCtrl and CameraCtrl exhibit significant deviations in camera following, indicating weak robustness in camera control. 2) Go-with-the-Flow often exhibits significant camera motion deviations and can sometimes result in collapsed or distorted video content. 3) GEN3C generates almost entirely static scenes where objects cannot move, resulting in rigid video content. Additionally, due to its reliance on 3D feature modeling, it is prone to scene penetration issues. In contrast, our method demonstrates superior performance in OOD scenarios, excelling in camera control accuracy, content plausibility, and motion dynamics.

**Comparison of longer video results.** Fig. 9 presents qualitative results for videos of different temporal lengths generated under the same scene conditions. As the video length increases, our method consistently maintains stable motion patterns and coherent temporal consistency across frames, without introducing noticeable drift or temporal artifacts. This indicates that the learned camera motion distribution generalizes well beyond the training horizon. Importantly, since the proposed CameraNoise is introduced solely at the noise modeling and conditioning stage during training, it does not modify the

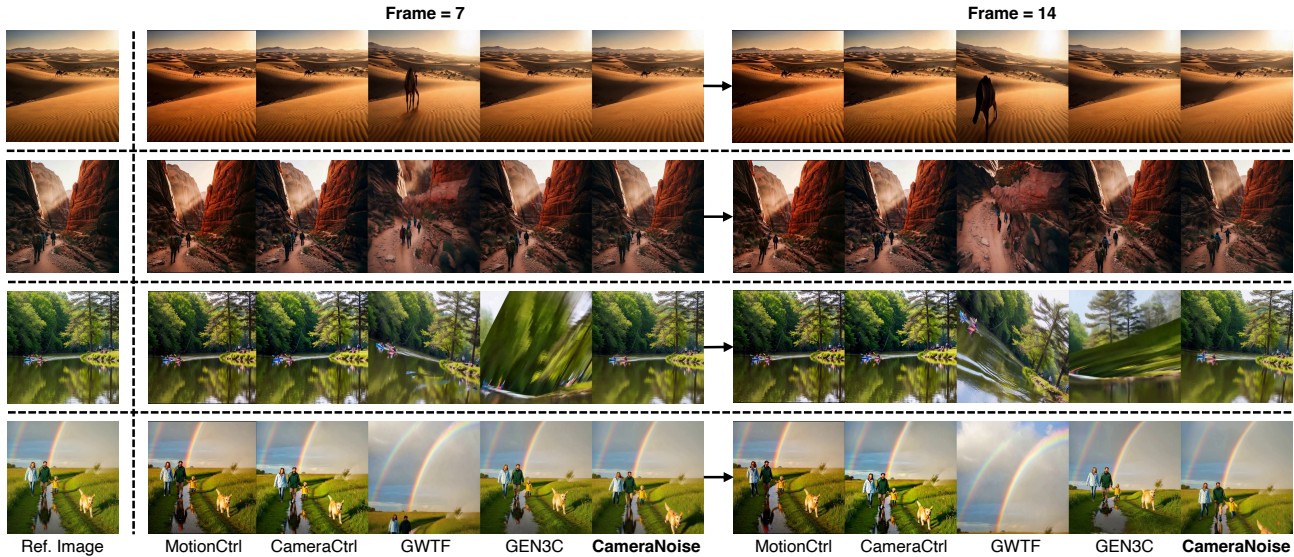

*Figure 8.* Qualitative results of CameraCtrl (He et al., 2025a), MotionCtrl (Wang et al., 2024b), Go-with-the-Flow (Burgert et al., 2025), and GEN3C (Ren et al., 2025) in dynamic OOD scenarios with image-to-video generation. In these four scenes, the type of the input camera poses are: move-left shot, move-up shot, move-clockwise shot, and move-down shot.

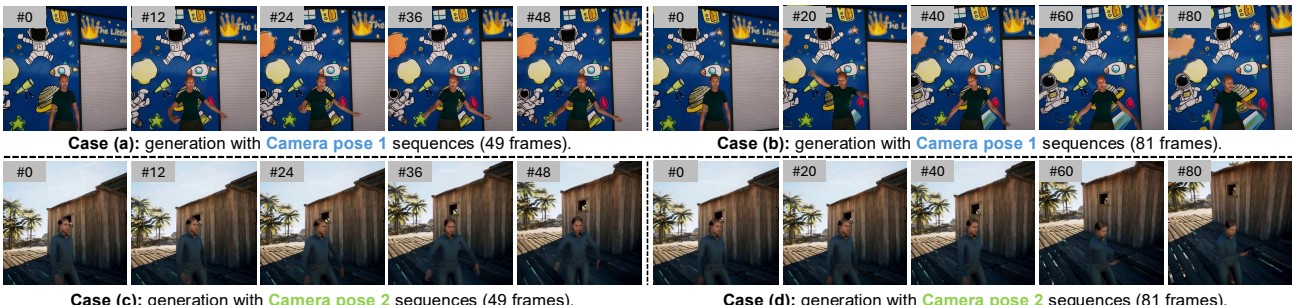

*Figure 9.* Stable results for longer videos on MultiCamVideo. We use the same camera sequences (camera poses 1 and 2) to generate videos of 49 (cases a and c) and 81 (cases b and d) frames, respectively. Cases (a-b) and cases (c-d) correspond to results generated from the same input images.

underlying architecture of the base diffusion model. As a result, the trained model fully preserves the long-range video synthesis capability enabled by the 3D RoPE positional encoding. In practice, this allows a model trained on 49-frame sequences to be directly extended to generate significantly longer videos, such as 81-frame sequences, which are typical target lengths for DiT-based video generation models.

**Qualitative evaluation under diverse camera motions.** We further conduct a qualitative evaluation of video generation under different camera motion conditions within the same scene, as shown in Fig. 10. Specifically, while keeping the scene content and reference image fixed, we apply four representative camera motion patterns to each reference image to drive the camera trajectories, including "moving left smoothly", "moving downward", "panning left", and "panning left and forward". From the visual results, we observe that under all camera motion settings, the model consistently generates videos that preserve high fidelity in terms of spatial structure, scene layout, and visual semantics, while accurately reflecting the intended camera motion trends. Notably, under cross-camera settings, the model maintains strong scene stability while flexibly responding to diverse camera motion instructions, without exhibiting noticeable viewpoint misalignment or temporal discontinuities. These results demonstrate that our method exhibits strong cross-camera generalization capability, enabling stable and controllable video generation across multiple camera motion patterns within the same scene.

**Cross-scene camera motion transfer.** In addition to generating camera trajectories from textual or visual conditions, our method can directly leverage camera parameters estimated from existing videos as driving signals. Specifically, we first recover the camera parameters from a source video using VGGT (Wang et al., 2025) and transform them into CameraNoise. It can then be seamlessly transferred and applied to other target scenes, enabling camera motion migration from a source

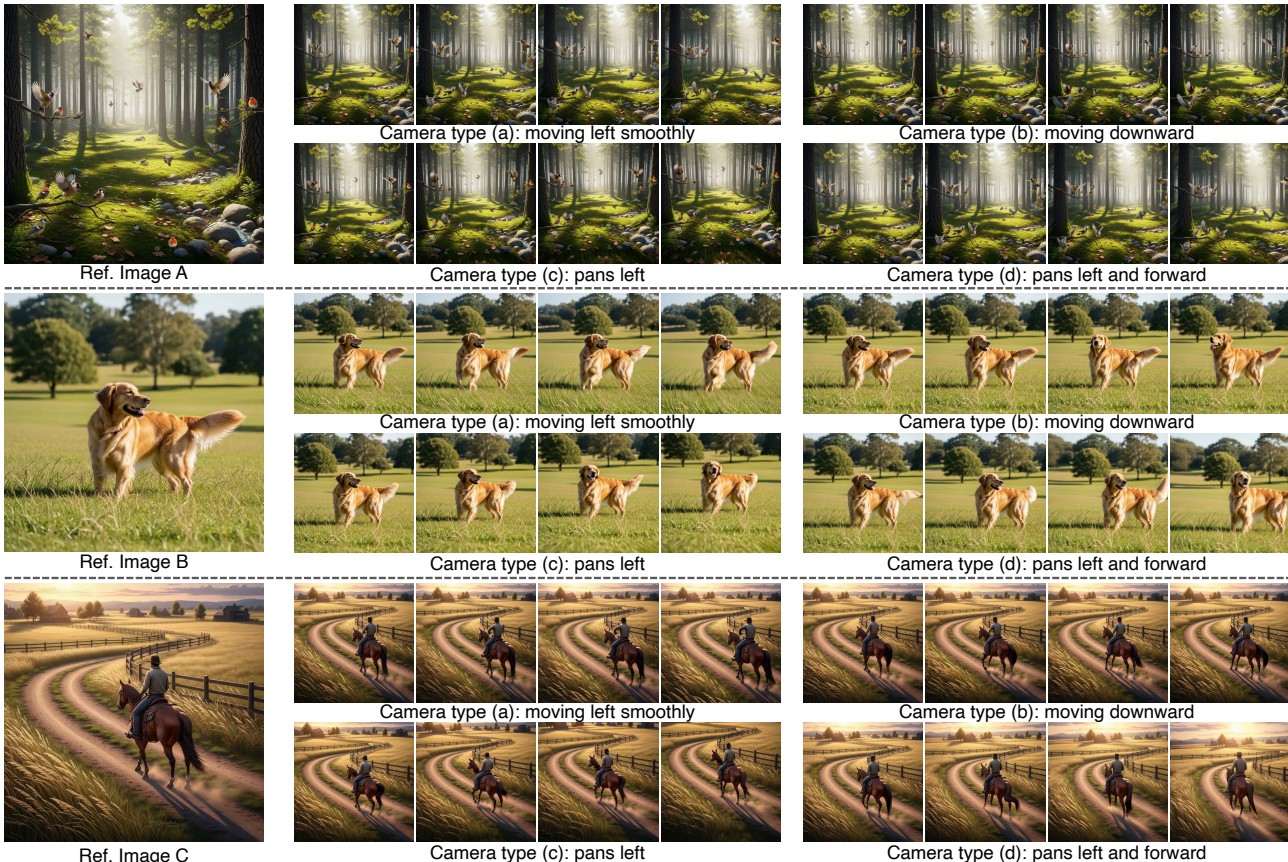

*Figure 10.* Qualitative results of video generation under different camera motion patterns within the same scene.

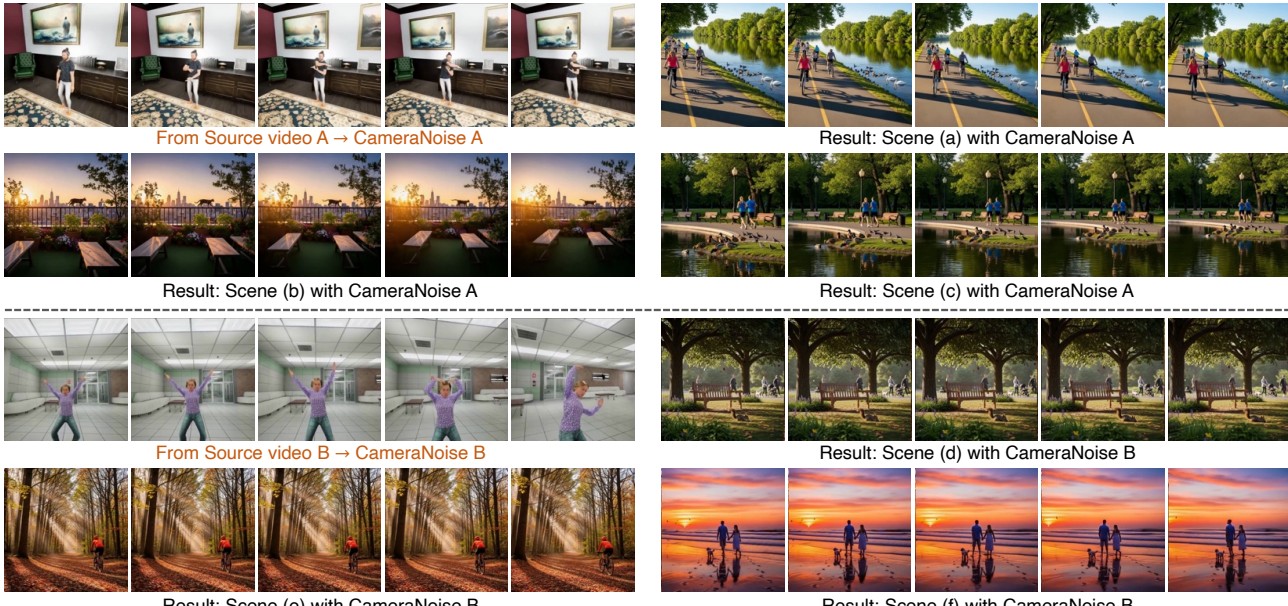

*Figure 11.* Cross-scene camera motion transfer via CameraNoise. Camera parameters are extracted from video A and video B and converted into CameraNoise A and CameraNoise B, respectively. The CameraNoise is then transferred to six different target scenes, shown in (a)–(c) and (d)–(f), to drive camera trajectories for video generation.

video camera to a target video camera. As illustrated in Fig. 11, we extract camera parameters from video A and video B and convert them into CameraNoise A and CameraNoise B, respectively. These CameraNoise representations are subsequently transferred to six different target scenes, shown in (a)–(c) and (d)–(f), to drive camera trajectories and generate the corresponding videos. Despite significant changes in scene content, the model consistently reproduces the camera motion patterns from the source videos while preserving the spatial structure and visual semantics of the target scenes. These results demonstrate that the proposed CameraNoise transfer mechanism exhibits strong cross-scene generalization. By decoupling camera motion from scene appearance, our approach enables effective reuse of rich camera motion priors embedded in real-world videos, substantially broadening the applicability of camera-controllable video generation and allowing more flexible deployment across diverse scenarios.

## B. Proof: Continuity of GRFlow.

In this section, we discuss our GRFlow algorithm, providing a formal proof of its continuous properties. We first define the continuous pixel domain $\Omega_c \subset \mathbb{R}^2$ and let the camera pose vary continuously in time $t \in [o, T]$: $\mathbf{E}(t) \in \text{SE}(3)$. Then, the continuous flow is:

$$\Phi : \Omega_c \times [0, T] \to \Omega_c, \quad \Phi(x, y, t) = \pi(\mathbf{E}(t) \cdot \ell(x, y)), \tag{15}$$

where $\ell$ is the lifting map and $\pi$ is projection to 2D. In practice, we sample pixels on a discrete grid $\Omega \subset \Omega_c$ and time is discretized into frame $t_i$. So, we can compute the GRFlow as:

$$\mathcal{G}_r^{i \to i+1}(x, y) = \pi(\Delta \mathbf{E}_i \cdot \ell(x, y)), \quad (x, y) \in \Omega, \tag{16}$$

with $\Delta \mathbf{E}_i = \mathbf{E}_i^\top \times \mathbf{E}_{i+1}$. As pixel resolution increases $D \to \infty$ and frame interval $\Delta t \to 0$, the discrete flow $\mathcal{G}_r$ converges pointwise to the continuous map $\Phi$.

$$\lim_{|\Omega| \to \infty, \Delta t \to 0} \mathcal{G}_r^{i \to i+1}(x, y) = \Phi(x, y, t_i), \quad \forall (x, y) \in \Omega_c. \tag{17}$$

Because we smooth the $\Delta \mathbf{E}$ during reprojection, the map is smooth. So, $\mathcal{G}_r$ is effectively a finite-sample approximation of a continuous flow on the 2D manifold, *i.e.,* $\mathcal{G}_r \approx \Phi|_\Omega$. Furthermore, since continuous coordinates are sampled on the discrete grid $\Omega$, sub-pixel displacements are typically approximated or interpolated, introducing minor errors. However, in our experiments, we find these errors to be negligible and they do not affect the results. Time Complexity of $\mathcal{G}_r$: furthermore, to construct GRFlow, we initialize a $D \times D$ grid. For each pixel in the grid, we compute $d \times \mathbf{K}$, where $\mathbf{K} \in \mathbb{R}^{3 \times 3}$ is the intrinsic matrix, resulting in a computational cost of $\mathcal{O}(D^2)$. The subsequent transformation using $\Delta \mathbf{E}$ also incurs $\mathcal{O}(D^2)$ operations. Hence, the overall time complexity of our algorithm is $\mathcal{O}(D^2)$ per frame.

## C. CameraNoise: One-to-One Mapping from Camera Pose via GRFlow.

### C.1. Why Direct Noise Interpolation Disrupts the Gaussian Prior?

In our framework, we consider a noise $N \sim \mathcal{N}(0, \sigma^2)$, where each pixel is independently drawn from a Gaussian distribution with zero mean and variance $\sigma^2$. This independent Gaussian prior is a fundamental assumption in diffusion-based generative models, ensuring that the denoising process operates on ideal white noise. When performing warping of the noise map using GRFlow, the target pixel coordinates often fall between discrete grid points, requiring interpolation (such as bilinear or bicubic) to obtain the warped noise values:

$$N'(x', y') = \sum_{i,j} w_{ij} N(x_i, y_j), \tag{18}$$

where $w_{ij}$ are interpolation weights satisfying $\sum_{i,j} w_{ij} = 1$. However, such interpolation inherently modifies the statistical properties of the original noise. Specifically, the output pixel $N'(x', y')$ is a weighted sum of neighboring noise values, which leads to a variance of:

$$\text{Var}(N') = \sum_{i,j} w_{ij}^2 \sigma^2 < \sigma^2. \tag{19}$$

This variance reduction diminishes the noise amplitude, thereby violating the original Gaussian prior. Moreover, after interpolation, adjacent pixels in the warped noise map receive contributions from overlapping input pixels, which introduces spatial dependencies. Consequently, the original assumption of independence in the Gaussian prior is no longer valid.

### C.2. Proof: One-to-One Mapping Among Camera Pose, GRFlow, and CameraNoise.

Suppose we have the camera parameters $\mathbf{K} \in \mathbb{R}^{3 \times 3}, \mathbf{E} \in \text{SE}(3)$, for a given frame of a video. Under the standard camera model, a 3D world point $X \in \mathbb{R}^3$ projects to a pixel coordinate $x \in \mathbb{R}^2$:

$$x \sim \pi(\mathbf{K}, \mathbf{E}, X) = K[\mathbf{R}|\mathbf{t}]X. \tag{20}$$

Our GRFlow is defined as the displacement field computed from the changes in camera matrices between consecutive frames:

$$\mathcal{G}_r(x) = f(\mathbf{K}_t, \mathbf{E}_t, \mathbf{K}_{t+1}, \mathbf{E}_{t+1}). \tag{21}$$

Meanwhile, the warped CameraNoise $\rho_{t+1}$ corresponds to the advected noise field obtained by solving the PDE:

$$\rho_{t+1} = \text{Advect}(\rho_t, \mathcal{G}_r). \tag{22}$$

To establish that the warped noise and the camera parameters form a one-to-one and unique mapping, *i.e.,* $\mid \rho_{t+1} \leftrightarrow (\mathbf{K}_{t+1}, \mathbf{E}_{t+1})$, we assume the following conditions:

- Optical invertibility: The camera projection is a single-valued mapping, *i.e.,* different 3D points do not project to the same pixel, and the projection is locally invertible.

- Uniqueness of GRFlow: The displacement field generated from continuous camera parameters uniquely determines the advect vector for each pixel.

- Uniqueness of the PDE: Given initial conditions and a smooth velocity field, the PDE admits a unique solution.

Under these assumptions, the mapping from GRFlow to the PDE advected velocity field is unique. Specifically, for each pixel $x$, the GRFlow induced by the camera parameter transformation is uniquely determined:

$$v(x) = \mathcal{G}_r(x) = \pi^{-1}(\mathbf{K}_t, \mathbf{E}_t, x) - \pi^{-1}(\mathbf{K}_{t+1}, \mathbf{E}_{t+1}, x). \tag{23}$$

Formally, assume there exist two distinct sets of camera parameters $(\mathbf{K}', \mathbf{E}') \neq (\mathbf{K}_{t+1}, \mathbf{E}_{t+1})$ that yield the same GRFlow for every pixel $x$. This contradicts the local invertibility of the camera projection at the pixel level, which ensures that no two different 3D configurations can project identically. Therefore, each GRFlow corresponds to a unique set of camera parameters, establishing a one-to-one correspondence.

Therefore, since the GRFlow is uniquely determined by the camera parameters, and the PDE solution is uniquely determined by the GRFlow and the initial noise, there exists a one-to-one correspondence between CameraNoise and the camera parameters as:

$$(\mathbf{K}_{t+1}, \mathbf{E}_{t+1}) \xrightarrow{\text{GRFlow}} \mathcal{G}_r \xrightarrow{\text{PDE Advect}} \text{CameraNoise}. \tag{24}$$

## D. Jacobian Matrix Defined in CameraNoise Warping.

In our CameraNoise warping algorithm, we employ an area scaling factor using the *Jacobian* matrix, which can correct the noise and ensure statistical consistency throughout the warping. We initialize the 2D Gaussian noise and warp it with the GRFlow as: $G(x)' = f(G(x)) = x + \mathcal{G}_r(x); x = (u, v)$, where $f$ denotes the mapping function. Within a local patch, $f(x)$ of pixel coordinates is approximated using a first-order Taylor expansion, yielding:

$$f(x + \Delta x) \approx f(x) + \mathcal{J}(x) \times \Delta x, \mathcal{J}(x) = I + \nabla d(x), \tag{25}$$

where $\nabla d(x)$ represents the *Jacobian* matrix of the mapping function $f(x)$ at $x$. It describes local linear transformation properties such as scaling, rotation, and shearing. To solve the $\mathcal{J}(x)$, we reuse the corresponding camera pose $\mathbf{E}$. We assume that:

$$p = K^{-1}\tilde{x} = \begin{bmatrix} (u - c_x)/f_x \\ (v - c_y)/f_y \\ 1 \end{bmatrix}, \quad X = d \times p, \tag{26}$$

where $d$ denotes the depth. Then, we get $X' = \mathbf{R} \times X + \mathbf{t}$ and project it onto the image plane $(u', v')$ with respect to the original pixel coordinates $(u, v)$ as:

$$u' = f_x \frac{X'_x}{X'_z} + c_x, \quad v' = f_y \frac{X'_y}{X'_z} + c_y. \tag{27}$$

For $\partial u'/\partial u$, using the quotient rule, we have:

$$\frac{\partial u'}{\partial u} = f_x \frac{(\partial_u X'_x)X'_z - X'_x(\partial_u X'_z)}{(X'_z)^2}. \tag{28}$$

The other three partial derivatives $[\partial u'/\partial v; \partial v'/\partial u; \partial v'/\partial u]$ are also computed in a similar manner. These partial derivatives are then combined to form the $\mathcal{J}$. Finally, the local area scaling factor is given by the absolute value of the determinant of $\mathcal{J}$ as $s(x) = |\det\mathcal{J}|$. Expansion occurs when the local volume increases ($\det J(x) > 1$), causing a single source pixel to split into multiple target pixels. In the bipartite graph, this results in one-to-many edges, with weights scaled by the local flow density. Contraction occurs when the local volume decreases ($\det J(x) < 1$), such that multiple source pixels map to a smaller region. In the bipartite graph, this corresponds to many-to-one edges, with missing regions filled using noise from the backward flow.

## E. Additional Implementation Detail.

**Evaluation metric details.** To assess camera motion, we employ the *TransErr* and *RotErr* (He et al., 2025a) metrics. RotErr measures rotational consistency of the camera using $\mathbf{R}$ matrices, while TransErr quantifies translation error as the $L2$ distance of the translation vectors $\mathbf{t}$:

$$\text{RotErr} = \sum_{j=1}^{m}(\sum_{i=1}^{n} \arccos \frac{tr(\mathbf{R}_{gen}^i \mathbf{R}_{gt}^{i\mathrm{T}})) - 1}{2})/m, \tag{29}$$

$$\text{TransErr} = \sum_{j=1}^{m}(\sum_{j=1}^{n} \|\mathbf{t}_{gt}^i - \mathbf{t}_{gen}^i\|_2^2)/m. \tag{30}$$

For $m$ videos, we compute the average over the $n$ frames of each video, and take the mean across all $m$ videos. To improve evaluation accuracy, we re-estimate camera poses for both ground-truth and generated videos using the advanced camera estimation model VGGT (Wang et al., 2025).

**Implementation details.** Our experimental details are divided into two parts: 1) CameraNoise generation: Experiments were conducted on a single NVIDIA GPU. We used the VGGT model to estimate camera parameters for data without pose annotations and generated GRFlow and CameraNoise using our proposed algorithm. Our method is **real-time capable**, processing videos at approximately 10 frames per second. 2) Integration of CameraNoise into video diffusion models: Experiments were conducted on 32 NVIDIA GPUs for model fine-tuning. We adopted the mainstream DiT Wan 2.1 model (Wan et al., 2025) as our training framework. CameraNoise was injected at the noise level, and the model was trained on the RealEstate10K training set using a LoRA-based training approach. We train videos with 49 frames and $1024 \times 576$ resolution. We set the LoRA rank to 32 and trained the model with a learning rate of 1e-4 and a batch size of 32.

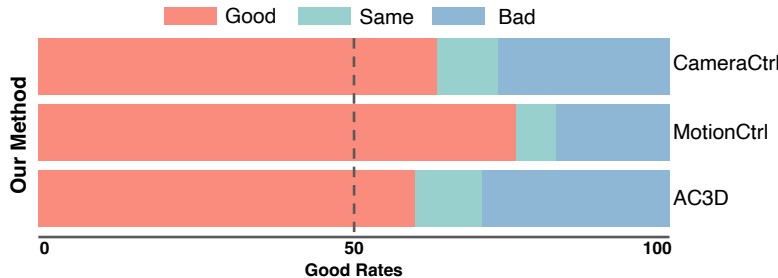

*Figure 12.* User study on 30 videos. Longer pink bars indicate higher satisfaction with our method.

## F. User Study

Fig. 12 presents the results of our user study evaluating the method with CameraCtrl (He et al., 2025a), MotionCtrl (Wang et al., 2024b), and AC3D (Bahmani et al., 2025) models, from a human perception perspective. In the study, participants were asked to compare our approach with baseline methods in terms of camera control and visual appeal. The results show that participants consistently preferred our method, indicating that it produces outputs that are not only more realistic but also more visually compelling, which also highlights its advantages over existing camera-controllable methods.

*Table 8.* Comparison of runtime overhead between our method and the Go-with-the-Flow (GWTF) (Burgert et al., 2025). We compare the runtime when processing an 81-frame video. The upper part reports the generation time of optical flow and GRFlow, while the lower part shows the runtime for optical-flow-based warped-noise generation and the runtime of our GRFlow-to-CameraNoise generation.

| Methods | Total time cost (seconds) ↓ | Per frame time cost (seconds) ↓ |
|---|---|---|
| Optical-flow generation (GWTF) | 4.455 | 0.055 |
| **GRFlow generation (Ours)** | **1.476** ($\times 3.02$) | **0.018** |
| Optical-flow to Warped-Noise (GWTF) | 9.558 | 0.118 |
| **GRFlow-to-CameraNoise (Ours)** | **6.802** ($\times 1.41$) | **0.084** |

## G. Time Cost.

**Time cost of GRFlow and CameraNoise generation.** Table 8 reports the generation time of our GRFlow method and the GRFlow-to-CameraNoise generation. We evaluate our method and the Go-with-the-Flow (GWTF) (Burgert et al., 2025) on an 81-frame video. In the first stage (from camera pose to GRFlow), our approach requires only *0.018 seconds per frame*, whereas optical-flow–based methods are $3.02\times$ slower. In the second stage (from GRFlow to CameraNoise), our synthesis process takes just *0.084 seconds per frame*, achieving a $1.41\times$ speed-up compared to the optical–flow–based approach GWTF. Therefore, our method can achieve a processing speed of *9.8 frames per second* from the initial camera pose to CameraNoise.

**Time cost of mode training and inference.** Our model is trained based on the Wan2.1 model. Since we do not modify the base model architecture, the training and inference times are roughly the same as the base model. On NVIDIA A100 hardware, we train the model using 32 GPUs with a total batch size of 32, a training resolution of $1024 \times 576$, and video samples of 49 frames. Each training step takes approximately 55.3 seconds. During inference, generating a 49-frame video at $1024 \times 576$ resolution with 25 steps takes about 390 seconds. When the frame count is increased to 81, the inference time rises to 780 seconds.

## H. More Qualitative Comparison.

Additionally, we provide more qualitative comparisons, including text-to-video generation results in Fig. 13 and Fig. 14, and image-to-video generation results in Fig. 15 and Fig. 16. These output videos are all conditioned by the input camera pose via our CameraNoise algorithm.

## I. Evaluation Data in RealEstate10K.

The IDs of the RealEstate10K test data used in our experiments are listed as follows:

| | | | |
|---|---|---|---|
| '0e512d350465a63c', | '55c99f9550523caa', | '6da3a9910fddd89e', | 'dd1c1d26525a2a1b', |
| 'e7020d449f50c737', | 'b3944d7eaf4f122b', | 'b5cc912bdce6fabd', | 'b859c7a7329c17b7', |
| 'b16d261e94f32108', | '82a317cb88729fe1', | '021575237abe0684', | '4881a65d7476d6dd', |
| '79bf72c63958d26a', | 'deb368fb90770550', | '1bf668db0194cf83', | '33f1be3a9ccf4e4b', |
| 'db2a88aeac0528ef', | '7526c18f191bcc1a', | '9d3b223e43672fb9', | '58c701594649d4e3', |
| 'c45cab04c3b22166', | '560e521aa9e864da', | 'bc70abfcfd247074', | '4242fb49c775710c', |
| '3b7443b24830d388', | '00cf0a94235771bb', | 'bb181e68415b169e', | '3e3d858083d20eab', |
| '0a7c052273895bb3', | 'bd3728fa823e6eb9', | '9b765910ee6573ec', | '446626a2bd617d24', |
| 'd5c6ad22b14eccef', | '7fad45df233ddce8', | '94e35563d865a2d6', | 'c35d057c102fe5ae', |
| '4daf919100e878ac', | '6986840ead0c9e9d', | '55e902a2cd2e976a', | '114d9c301b847239', |
| 'd35f508ebd80e610', | 'b8fda11b15ac85ff', | '91234df26c87a72b', | '86e00a902518e491', |
| '3018aa8ad3eb5dca', | 'bb6fa5bdafc14e8c', | '5d9f7f0205f7bee5', | '511ad5dc10db6932', |
| '6f243139ca86b4e5', | 'dd5288bacc7da7cf', | '23099812f662b3ec', | '188e6f96fa74ebe7', |
| '3fb3327a177a0175', | '46e0654ccb5d88cf', | '1b09ad5460c05077', | 'bf756257ffdd0017', |
| 'b43aa92e530e2aa5', | '14cf1f92ca13d605', | '122cb7d5ea4a99df', | '6beedb01303bb667', |
| '1b2937e192040745', | 'a3bc75f0a32b1501', | '8317ac8848eb60de', | '645cc7949386d427', |
| '861e80a1959788a7', | 'b2fa3530dcd34093', | '93a8bf0ecd7eafcf', | 'c99afff025000694', |
| '2f7f2369486cc959', | '6ee670df48229b4e', | 'f1d9d9caf8269fe6', | '330b925cef643b3f', |
| '5151d3969e328df1', | '5a15212752d1659a', | 'c8c2d887b38f6dc5', | '3e68931874661724', |
| 'eac593ae8fff36a3', | '7c7bc5285126e6ad', | '504e7ca0f7bb3427', | 'a6bc234b1ce6ca2b', |
| '8f04f919b046336c', | '4c69bf407b142b93', | 'a1d38185b8f59a4b', | 'a8baf24bbae943a0', |
| '0ed8b86b87a30d38', | '5bb9c1498799204b', | '02b59cd60efb924e', | 'd6b3d15ff42247fb' |
| '83ceef672f798063', | 'ce8bc5948d0dd3f3', | 'edbdb6ad0b956efb', | |
| 'a552d52c34c2c920', | 'c3c33ceed1308b42', | '0463d74358aca878', | |
| '0e8a52a174610350', | '0c916bcc9351521e', | '56ae4fff81255579', | |
| '4f0aa4ef8976dc1b', | 'ba604e2b0ab1be25', | '40f92f1e65a5e1dd', | |

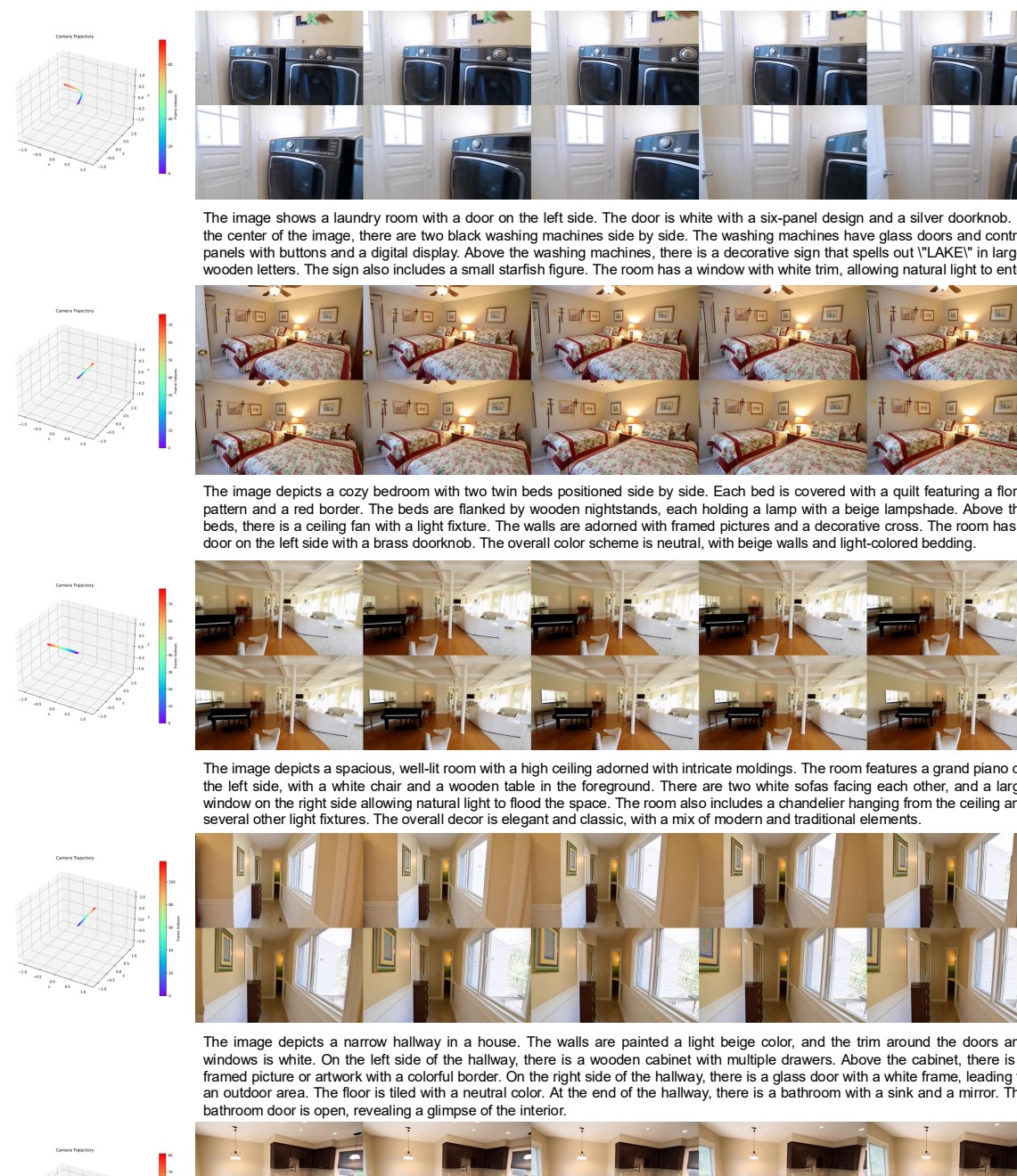

The image shows a laundry room with a door on the left side. The door is white with a six-panel design and a silver doorknob. In the center of the image, there are two black washing machines side by side. The washing machines have glass doors and control panels with buttons and a digital display. Above the washing machines, there is a decorative sign that spells out \"LAKE\" in large, wooden letters. The sign also includes a small starfish figure. The room has a window with white trim, allowing natural light to enter.

The image depicts a cozy bedroom with two twin beds positioned side by side. Each bed is covered with a quilt featuring a floral pattern and a red border. The beds are flanked by wooden nightstands, each holding a lamp with a beige lampshade. Above the beds, there is a ceiling fan with a light fixture. The walls are adorned with framed pictures and a decorative cross. The room has a door on the left side with a brass doorknob. The overall color scheme is neutral, with beige walls and light-colored bedding.

The image depicts a spacious, well-lit room with a high ceiling adorned with intricate moldings. The room features a grand piano on the left side, with a white chair and a wooden table in the foreground. There are two white sofas facing each other, and a large window on the right side allowing natural light to flood the space. The room also includes a chandelier hanging from the ceiling and several other light fixtures. The overall decor is elegant and classic, with a mix of modern and traditional elements.

The image depicts a narrow hallway in a house. The walls are painted a light beige color, and the trim around the doors and windows is white. On the left side of the hallway, there is a wooden cabinet with multiple drawers. Above the cabinet, there is a framed picture or artwork with a colorful border. On the right side of the hallway, there is a glass door with a white frame, leading to an outdoor area. The floor is tiled with a neutral color. At the end of the hallway, there is a bathroom with a sink and a mirror. The bathroom door is open, revealing a glimpse of the interior.

The image depicts a modern kitchen and dining area. The kitchen features dark wooden cabinets and a countertop with a light-colored backsplash. There is a double sink and a window above the sink. The dining area has a wooden table with two chairs, each set with a white plate, a fork, and a knife. The table is placed near a set of French doors that lead to an outdoor area, which appears to be a patio or deck. The walls are painted in a light beige color, and there is a pendant light hanging from the ceiling. The overall atmosphere of the space is clean and contemporary.

*Figure 13.* Qualitative results of text-to-video generation (1/2) for video sequences under complex camera poses in RealEstate10K dataset. For each case, the leftmost panel shows the camera trajectory used for control, with prompts provided below each image.

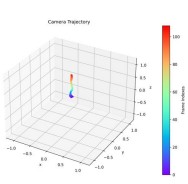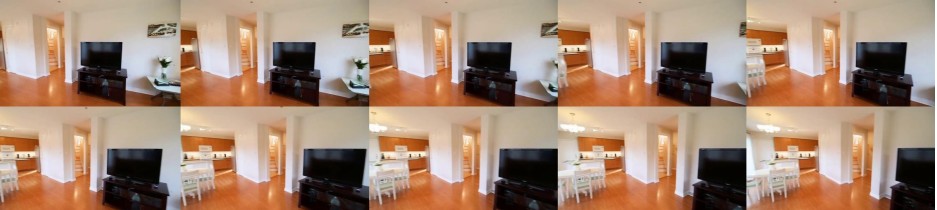

The image depicts a modern, well-lit living room with a hardwood floor. The room features a flat-screen television on a dark wooden stand, positioned against a wall. To the right of the television, there is a small table with a vase and some decorative items. The living room opens into a dining area on the left, with a white dining table and chairs. The kitchen is visible through an open doorway, with wooden cabinets and a white countertop. There is a staircase leading to an upper level on the right side of the image.

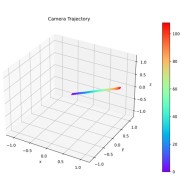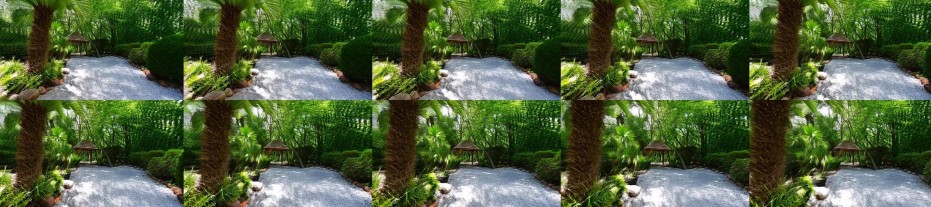

The image depicts a serene and lush garden setting. The foreground features a gravel pathway that leads towards a wooden structure, possibly a gazebo or a small pavilion. The pathway is bordered by a variety of plants and trees, including tall palm-like trees and smaller shrubs. The garden is well-maintained, with rocks and stones strategically placed along the pathway, adding to the natural aesthetic. The background is filled with dense foliage, creating a canopy of greenery that filters the sunlight, casting dappled shadows on the ground.

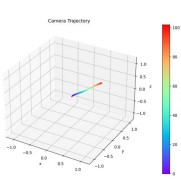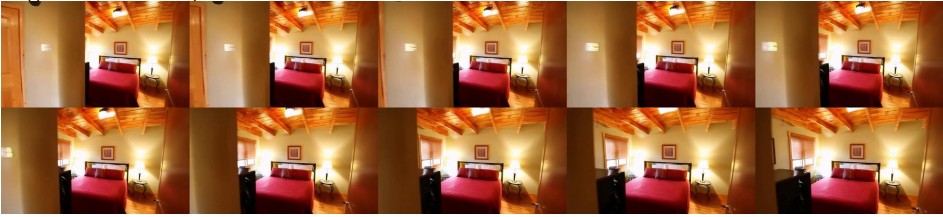

The image depicts a cozy bedroom with a warm and inviting atmosphere. The room features a wooden ceiling with exposed beams, adding a rustic charm to the space. The walls are painted in a light, neutral color, which complements the wooden elements. In the foreground, there is a bed with a red bedspread and several pillows. The bed is positioned against the wall, and a small table with a lamp is placed beside it. The lamp is turned on, casting a soft, warm light that enhances the cozy feel of the room.

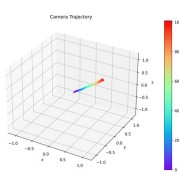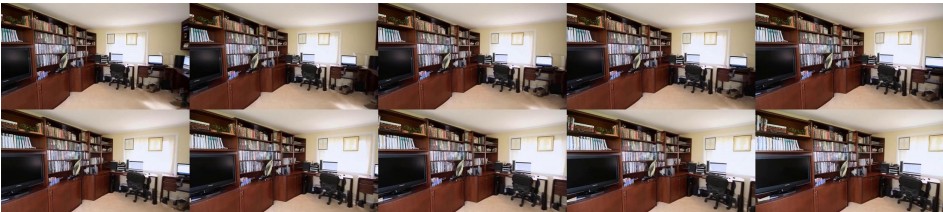

The image depicts a well-organized home office and entertainment area. The room features a large bookshelf filled with books and various items, including a television on the left side. There is a desk with a computer setup in the center of the room, and another desk with a computer setup on the right side. The desks are accompanied by chairs, one of which is a rolling office chair. The room also has a small plant and framed certificates or awards on the wall.

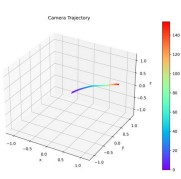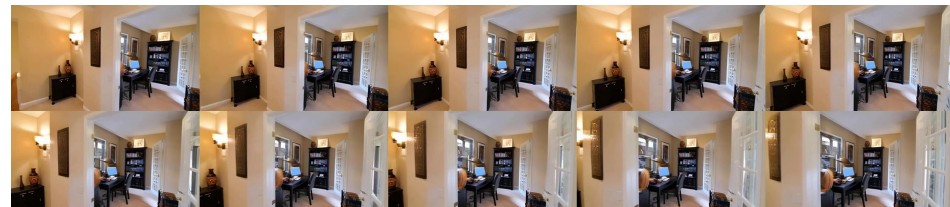

The image depicts an interior hallway leading to different rooms. On the left side, there is a staircase with a wooden railing. The hallway has beige walls and is illuminated by wall sconces. At the end of the hallway, there is a small black cabinet with a decorative vase on top. To the right, there is a white French door leading to another room. The room beyond the door has a dark bookshelf filled with books and various items, including framed pictures and a desk with a computer. The overall atmosphere of the space is warm and inviting.

*Figure 14.* Qualitative results of text-to-video generation (2/2) for video sequences under complex camera poses. For each case, the leftmost panel shows the camera trajectory used for control, with prompts provided below each image.

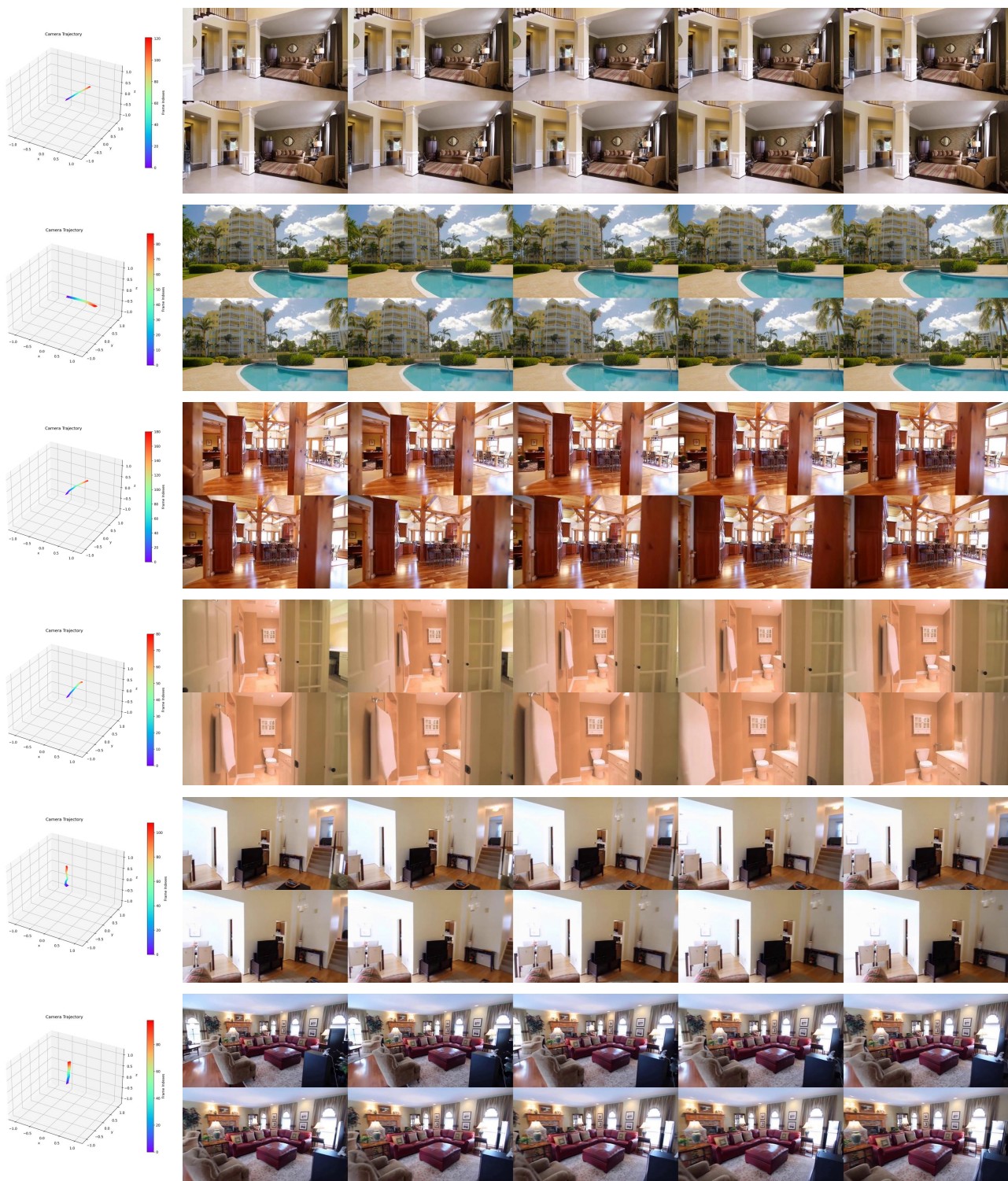

*Figure 15.* Qualitative results of image-to-video generation (1/2) for video sequences under complex camera poses. For each case, the leftmost panel shows the camera trajectory used for control, and the reference image corresponds to the first frame.

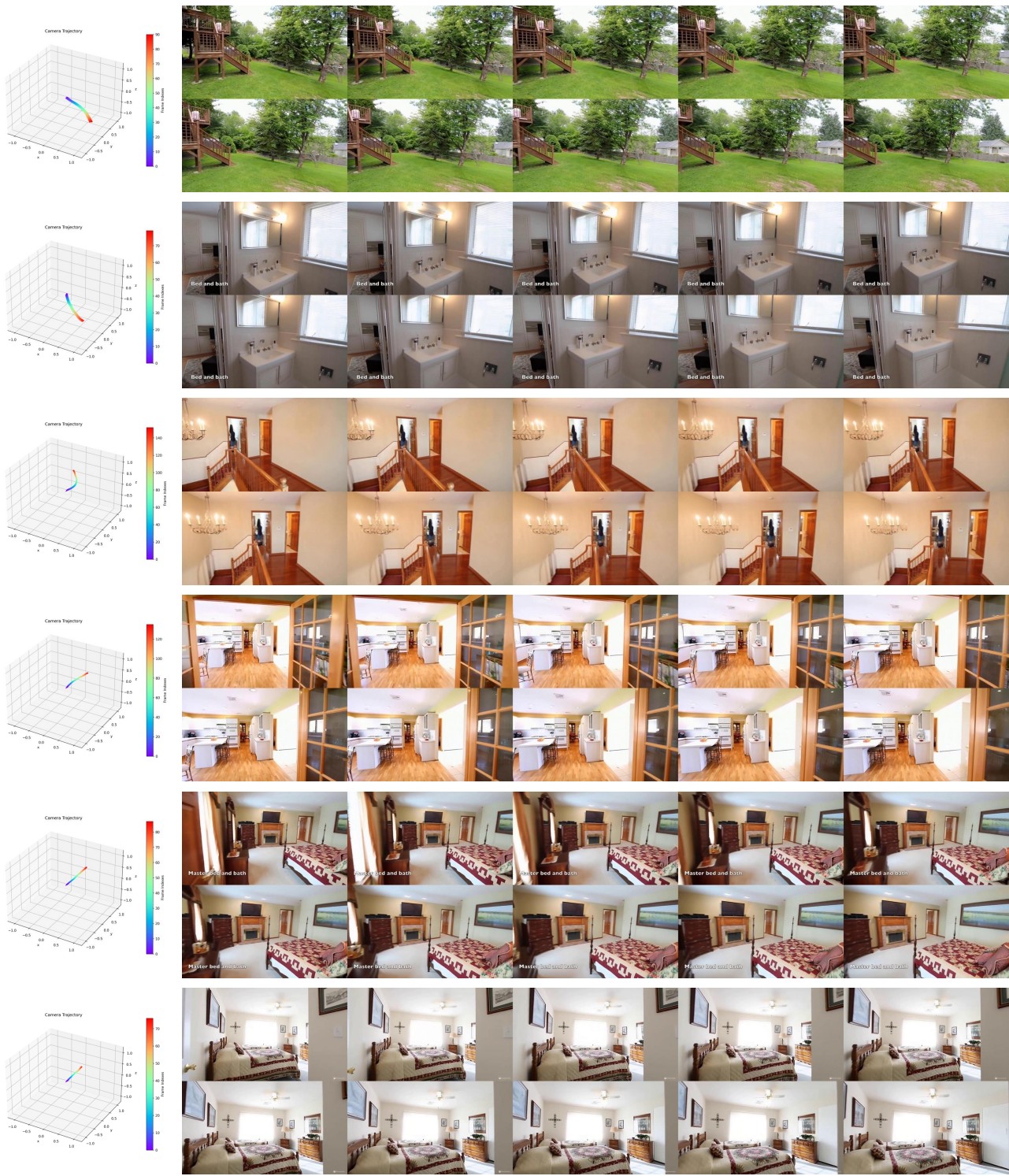

*Figure 16.* Qualitative results of image-to-video generation (2/2) for video sequences under complex camera poses. For each case, the leftmost panel shows the camera trajectory used for control, and the reference image corresponds to the first frame.

