# OpenReview forum: "CameraNoise: Enabling Faithful Camera Control in Video Diffusion through Geometry-Flow-Guided Noise Warping"
_ICML.cc/2026/Conference — ICML 2026 regular_

### Official Review · Reviewer_bgRv · 2026-02-21

**Soundness:** 3
**Presentation:** 2
**Significance:** 2
**Originality:** 3
**Overall Recommendation:** 4
**Confidence:** 5

**Summary:**

This paper proposes CameraNoise, a method for camera-controlled video generation. Previous works either inject camera parameters into the network or into the noise based on optical flow. To decouple motion from scene appearance, this work introduces a geometry-guided reprojection flow to guide the control. The warping is formulated as a partial differential equation, leading a one-to-one mapping from camera poses to CameraNoise.

**Compliance With Llm Reviewing Policy:**

Affirmed.

**Final Justification:**

My concerns are mostly addressed, though I think the dynamic generation results are not super convincing.

**Key Questions For Authors:**

The method seems reasonable but currently it is difficult to judge the work properly. I would like authors to comment on following:

- How does the method compare computationally with previous works?
- How well does the method work on dynamic scenes and out-of-distribution prompts beyond the fine-tuning data?

For now, I am negative, but I am happy to re-consider my score based on a rebuttal and other reviews.

**Limitations:**

Yes.

**Strengths And Weaknesses:**

Strengths:

- Reasonable approach: Decoupling motion from appearance based on geometry-guided flow makes sense and is reasonable.

Weaknesses:

- Missing computational costs: It is not clear how expensive is it to construct the guidance, especially compared to optical flow-based guidance as in previous works. The pipeline seems very complex, hence memory and time analysis would have been great.
- Base model not clarified: It is not clear from the paper what base model this work is using and if it is comparable in its capabilities to the baselines.
- Dynamic generation capabilities: The results seem to focus mainly on static scenes, the dynamic results are slow-motion and seem to be entangled with the speed of the camera.

---

> ### Author Rebuttal · Authors · 2026-03-31
>
> We thank Reviewer bgRv for the thoughtful and constructive comments. It is evident that the reviewer spent considerable time carefully reading our work and provided valuable insights and suggestions. We really appreciate that the reviewer thinks that our method is reasonable. We also thank you for the detailed feedback and the recognition of the strengths. Below, we address the specific points.
>
> > Q1: Missing computational costs.
>
> **Response:** We apologize for any confusion. Due to the page limit, we placed the comparison with optical-flow-based guidance in Appendix G. As shown there, our camera-pose-to-GRFlow module achieves a 3× speedup, and the GRFlow-to-CameraNoise step is 1.4× faster than the optical-flow-based method GWTF. For training and inference, the runtime is roughly the same as that of the base model Wan 2.1, indicating that our diffusion process introduces no obvious additional time cost compared with GWTF.
>
> In addition, in the rebuttal, we provide further time and memory analyses of the PDE solver; please refer to our response to Table A in Q2 for Reviewer TdXg. As shown there, the proposed PDE method introduces only limited time and memory overhead. Overall, our method is efficient and practical to use. We will also open-source our code and models.
>
> > Q2: Base model not clarified.
>
> **Response:** Thanks for your suggestion. Due to the page limit, we placed the implementation details in Appendix E. We indicate that we adopt the mainstream Wan 2.1 model as our base model in Line 933.
>
> Furthermore, we compare different camera injection methods under the same backbone. Please refer to our response to Table A in Q3 for Reviewer QUog for rebuttal. This ablation clearly demonstrates that the proposed CameraNoise is an effective way to inject camera pose into video diffusion models.
>
> > Q3: Dynamic generation capabilities.
>
> **Response:** Thanks for your suggestion. Due to the page limit, we placed the results on dynamic scenes and out-of-distribution scenarios (including prompts and reference images) in Fig. 6, Table 6, Fig. 7, and Fig. 11 of Appendix A. We also provide qualitative results in video format on our anonymous project page. These results show that our method performs well on these challenging scenes. In particular, Fig. 11 presents results on driving scenes, which are representative dynamic scenarios.
>
> In fact, to ensure that our method can handle such challenging cases, we mainly focus on the following aspects:
>
> - Accurate CameraNoise representation: The mapping from camera poses to CameraNoise is one-to-one, enabling our method to reliably handle a wide range of camera motions, from subtle movements to large transitions. As shown in Section 2 of our anonymous project page, the constructed CameraNoise effectively preserves continuous and dynamic camera motion patterns.
> - Preserving the diffusion backbone architecture: Our method introduces CameraNoise at the noise input stage without modifying the diffusion backbone, thereby incorporating camera parameters while preserving the model’s original generalization ability.
> - Reliable training data support: To improve dynamic generation in zero-shot scenarios, we deliberately choose DynPose-100K for training. As a high-quality dataset with strong coverage of dynamic scenes, it provides solid support for fine-tuning and leads to more natural and stable results.
>
> Furthermore, we would like to clarify that object motion is not entangled with camera speed. This can be clearly observed in Section 9 (“Dynamic and Outdoor Scenes”) of our anonymous project page, e.g., in the 3rd and 4th examples of the first row, the 1st example of the second row, and the 4th example of the third row. Across these cases, object motion is clearly decoupled from camera speed. For instance, in the 4th example of the third row, the camera moves downward slowly, while the dog shows a rapid head-turning motion. This demonstrates that object motion is not directly constrained by camera motion speed.
>
> In addition, the quantitative and qualitative results in out-of-distribution scenarios, as well as the cross-scene camera motion transfer results, were praised by Reviewer TdXg. Reviewer QUog also considered our qualitative examples to be reasonably representative, rather than merely cherry-picked videos.
>
> > Q4: Key questions: i.e., compare computationally with previous work and work on dynamic scenes and out-of-distribution prompts.
>
> **Response:** Thank you for this helpful suggestion and for your willingness to reconsider the score. We have addressed these two concerns in our responses to Q1 and Q3. In brief, our method introduces only modest runtime and memory overhead and is more efficient than prior optical-flow-based alternatives. We also provide both quantitative and qualitative evidence showing that it remains effective on dynamic scenes and out-of-distribution prompts beyond the fine-tuning data. We will make these points clearer in the revised version.

---

> > ### Author Rebuttal · Reviewer_bgRv · 2026-04-02
> >
> > Thanks for the response.
> >
> > I have remaining concerns over the dynamic scenes. Since the base model has been clarified to be Wan, I do not understand why the dynamic videos are only 2 seconds. Since Wan can produce longer videos. The camera motion in those results is very limited and the motion is a bit slow-motion. Could you please clarify?

---

> > > ### Author Response · Authors · 2026-04-03
> > >
> > > We thank Reviewer bgRv for the follow-up comments and for clearly outlining the remaining concerns. We appreciate the opportunity to further clarify these points and address them more directly below.
> > >
> > > > Q1: Long video generation.
> > >
> > > **Response:** Thanks for your suggestion. In the initial submission, we explicitly stated in **Line 375** that the long-video generation results are presented in **Fig. 8 of the Appendix**. At the same time, as a representative dynamic scenario, in **Line 377 and Line 746**, we provide long-video results in driving scenarios to verify the effectiveness of our method in dynamic environments. Besides, as shown on our anonymous project page, all of these driving videos are longer than 4 seconds, rather than 2 seconds. Since CameraNoise is injected in the noise space as a conditioning signal, it does not require any modification to the architecture of the Wan model. Therefore, during inference, the model can be directly extended to generate videos with 81 frames. The related explanation is also provided in **Lines 646–651** of the manuscript.
> > >
> > > We would also like to emphasize that the core contribution of this paper is the proposed novel warping method, CameraNoise, which enables the synthesis of temporally structured Gaussian noise from camera parameters and incorporates it into video diffusion for camera-controllable video generation. The long-video results serve to verify that our method naturally extends to longer sequences without modifying the video diffusion backbone, while also highlighting its broader applicability.
> > >
> > > Moreover, in this rebuttal, to further alleviate the reviewer’s concerns, we supplement the paper with additional validations under more challenging camera motions and dynamic scenes. Specifically, we introduce continuously varying camera trajectories and highly dynamic amusement-park scenes to assess the effectiveness and robustness of our method in dynamic scenarios, under complex camera motions, and in long-video generation settings. These results are available in **Rebuttal-A1** on our anonymous project page: https://lizaigc.github.io/. These results demonstrate the robustness of our method in dynamic scenarios and long-video generation settings.
> > >
> > > > Q2: The camera motion in those results is very limited and the motion is a bit slow-motion.
> > >
> > > **Response:** Thanks for your suggestion. The camera motions tested in our evaluation follow the settings adopted in prior works, including MotionCtrl, CameraCtrl, and Gen3C. In the experiments of this paper, we have extensively evaluated a wide range of typical camera motions, including translations in the up, down, left, and right directions, as well as rotational motions. These results can be found in **Fig. 6 through Fig. 11**. In the additional visualizations provided in **Rebuttal-A1** on our anonymous project page of this rebuttal, we deliberately selected camera motions with changing directions and rapid movement to further demonstrate the effectiveness of our method under complex camera motion settings.
> > >
> > > Furthermore, we also want to explain that the camera motion and object motion in our method are not always coupled. We acknowledge that, in certain specific scenarios, when object motion and camera motion occur simultaneously, they may appear to be coupled; a typical example is an onboard camera in driving scenes. However, in more general settings, object motion is not determined by camera motion, and can instead be deliberately controlled through text prompts.
> > >
> > > To demonstrate this, we provide corresponding visualizations in **Rebuttal-A2** on the anonymous project page. Specifically, we intentionally select scenes with a static camera, where it can be clearly observed that "the boat" in the scene is still able to move freely even when the camera remains fixed. **This directly shows that the camera motion and object motion are not coupled, and the object motion is not limited to slow motion.** The three cases are driven by different text prompts, such that the motion speed of the object can also be controlled by the text condition. This phenomenon can be clearly observed from the provided results.
> > >
> > > We hope that the above clarifications adequately address the reviewer’s remaining concerns.

---

### Official Review · Reviewer_Ru3T · 2026-03-12

**Soundness:** 3
**Presentation:** 3
**Significance:** 3
**Originality:** 3
**Overall Recommendation:** 5
**Confidence:** 4

**Summary:**

The paper is solving the issue of image generation with a given camera motion. The authors argue the necessity of injecting the camera parameters into the noise rather than the diffusion architecture in order to enforce geometrical consistency.

The method takes as input camera parameters $K_i, E_i$. A flow field is computed that maps coordinates in one image to coordinates in another image. A 2D grid is sampled and reprojected into the 3D scene using the camera parameters and constant pseudo depth $d$. It is then projected into the other image using the relative orientation and projection of the second camera. The result is a flow field based on the geometry of the scene rather than optical flow based matching of pixels. This flow field is used to transform the latent Gaussian image directly.
As a result two consecutive frames are related in their input noise by this displacement/warp field.
In order to avoid artifacts due to interpolation a linear advection PDE is solved to warp the discrete noise image while preserving its Gaussian properties.

VGGT is used for computing camera intrinsics and extrinsics.

**Compliance With Llm Reviewing Policy:**

Affirmed.

**Final Justification:**

My main concerns have been answered and I think it is an interesting and novel approach.

**Key Questions For Authors:**

- Since the depth $d$ is chosen as a constant the reprojection should describe a case were the first camera is observing the inside surface of a sphere. How exactly is $d$ chosen? Does the method avoid the parallax issues of other methods due estimating more global motion? Could you also approximate the warp field using a homography? This would be similar to $d\rightarrow \infty$ or using a depth map that is consistent with a plane at some fixed distance.

**Limitations:**

yes

**Strengths And Weaknesses:**

# Strength

The methods is very relevant for video generation with a consistent controllable camera motion.
The approach is overall sound and seems quite novel.  Using a warping field on the noise directly is a very interesting approach and should be further explored.

The metrics and evaluations also seem to show superior results.
On the project page we can see a demonstration of the videos, which helps the understand of the method


# Weaknesses

The are many mathematical inconsistencies and problems due to notation.
- Equation (2) and (3) seem inconsistent. In (2) a row vector is multiplied from the left and in (3) from the right.
- For example, $\Delta E = E_i^T \times E_{i+1}$ indicates a cross product. But it should describe a matrix multiplication.
However, even then $E=[R,t]$ should be a $3\times 4$ matrix. If $E$ is a $4\times 4$ a homogeneous matrix then formula for relative transformation should be $\Delta E = E_i^{-1} \cdot E_{i+1}$ or $\Delta E = E_i \cdot E_{i+1}^{-1}$ depending on the direction. (correct me if I'm wrong)
- The PDE method should be elaborated further

# Conclusion

Overall I believe the method is very interesting with a lot of potential. I would increase my ranking if the question and remarks are fixed.

---

> ### Author Rebuttal · Authors · 2026-03-31
>
> We thank Reviewer Ru3T for the thoughtful and constructive comments. It is evident that the reviewer spent considerable time carefully reading our work and provided valuable insights and suggestions. We really appreciate that the reviewer thinks that our method is very interesting with a lot of potential. We also thank you for the detailed feedback and the recognition of the strengths. Below, we address the specific points.
>
> > Q1: Equations (2) and (3) seem inconsistent.
>
> **Response:** Thank you for pointing this out. In the revised version, we will unify them using the same homogeneous column-vector convention. For Eq (2), we modify it to:
>
> $\mathbf{p}_ {x,y} = d \tilde{\Omega}_ {x,y}, \tilde{\Omega}_ {x,y} = \mathbf{K}_ i^{-1} \Omega_ {x,y} \in \mathbb{R}^3$.
>
> For Eq. (3), we modify it to:
>
> $\ell : \Omega \to \mathbb{R}^3, \quad \ell(x, y) = d \mathbf{K}_ i^{-1} [x, y, 1]^\top$.
>
> Besides, we will unify this notation throughout the revised manuscript.
>
> > Q2: Definition of the relative transformation.
>
> **Response:** Thank you for pointing this out and we apologize for the ambiguity. The symbol $\times$ was a notation error and was not intended to denote a cross product. This symbol will be removed globally where it was incorrectly used for matrix or scalar multiplications.
>
> Furthermore, we correctly define the world to camera matrix $E_ i$ as a $4 \times 4$ homogeneous matrix in $SE(3)$ and define the relative transformation as $\Delta E_ {i\rightarrow i+1}=E_ {i+1}E^{-1}_ i$. The reprojection process is now explicitly written as:
>
> $\tilde{\mathbf{p}}'_ {x,y}=\Delta E_ {i\rightarrow i+1}\tilde{\mathbf{p}}_{x,y}$
>
> To ensure mathematical rigor. We will correct the reprojection formula to accurately reflect the mapping from 3D homogeneous coordinates to the 2D image plane. These contents will be updated in the revised version.
>
> > Q3: The PDE method should be elaborated further.
>
> **Response:** We thank the reviewer for this suggestion. The PDE method is a core mechanism for transforming camera poses into CameraNoise. We have elaborated on its details as follows:
>
> -	The purpose: The PDE method bridges camera geometry and noise evolution. It ensures that the initial noise “flows” consistently with the camera trajectory, enabling stable video generation.
> -	How to define the PDE: We formulate noise propagation as a linear advection PDE (Eq. 8), which we solve on a discrete grid using a bipartite graph (Eq. 9). Noise values are treated as “mass” flowing along the vectors of GRFlow.
> -	The results: We employ the PDE to generate CameraNoise, which is a noise sequence that is spatio-temporally aligned with camera poses while maintaining a bijective mapping.
> -	Advantages of the proposed PDE: Unlike naive warping, our PDE-based method preserves the Gaussian prior by preventing variance blurring and handles density changes (e.g., during zooming) through Jacobian-aware normalization (the denominator in Eq. 9), thereby ensuring a uniform noise field without artifacts.
> -	Time and memory costs: We provide a detailed analysis of the runtime and memory overhead of the PDE method. Since this concern is closely related to another reviewer’s question, we refer the reviewer to our response to Table A in Q2 for Reviewer TdXg. As shown there, the proposed PDE method introduces only modest overhead. For a 1024 × 576 video, generating CameraNoise runs at 20.17 FPS and uses 644.95 MB of memory, yielding a 1.4× speed-up over the optical-flow-based approach GWTF. These results further support the practicality of our method for large-scale real-world deployment, and we will clarify this point in the revision.
>
> > Q4: Key questions, e.g., how exactly is $d$ chosen, whether the method avoids parallax issues, and whether the warp field can be approximated using a homography.
>
> **Response:** Thank you for this insightful question. We agree that a constant $d$ does not represent the true scene depth distribution. In our method, $d$ is used as a fixed pseudo-depth reference to convert relative camera motion into a geometry-guided global motion field for CameraNoise construction, rather than to recover accurate 3D geometry. We use a fixed hyper-parameter ($d$=0.5 in our experiments), which works robustly across diverse scenes.
>
> Furthermore, our method does not aim to explicitly solve parallax through depth estimation. Instead, it provides a camera-induced global motion prior, while the detailed scene geometry and parallax effects are still modeled by the diffusion backbone during denoising. We agree that homograph is a related approximation under a planar-scene assumption or the large-depth limit, but our method is not a single-plane warp. Rather, it uses relative camera motion with a finite pseudo-depth to construct a practical camera-aware noise prior for controllable video generation.
>
> Thank you again for these helpful suggestions and for your willingness to increase the score.

---

> > ### Author Rebuttal · Reviewer_Ru3T · 2026-04-02
> >
> > I thank the authors for the insightful answers and for adjusting the notation.
> >
> > Ideally I would like to see some exploration on the parameter $d$. Specifically, how much bias it adds to the noise. How much disentanglement of the parallax and movement does the diffusion do and how much is added by the prior. Because if it is done by the diffusion model a planar-homography might be also enough as a prior.
> >
> > I am still improving my score already.

---

> > > ### Author Response · Authors · 2026-04-03
> > >
> > > We sincerely thank Reviewer Ru3T for the continued thoughtful feedback and for the positive reassessment of our work. We greatly appreciate the reviewer’s recognition of our clarifications and the increased score, as well as the additional questions that help further improve the paper. Below, we respond to these points in detail.
> > >
> > > > Question: Exploration on the parameter $d$.
> > >
> > > **Response:** Thank you for your thoughtful comment. To further address the reviewer’s question regarding the role of parameter $d$, we provide additional visualizations in this rebuttal to more directly analyze its effect on both the CameraNoise representation and the final video generation results. Specifically, in **Rebuttal-B** on our anonymous project page, i.e., https://lizaigc.github.io/, we present visualizations of CameraNoise generated under different values of $d$ across multiple scenes (video format). We observe that when $d$ is small, e.g., ($d=0.1$), small motions in the original video are significantly amplified. In contrast, as $d$ increases, the push-in and pull-out effects encoded in CameraNoise become substantially weakened, and rotational camera motions also exhibit visible artifacts. This is because, although $d$ is a pseudo-depth parameter rather than the true physical depth of the scene, it still represents the relative distance between the camera and the reference plane, and therefore directly affects the displacement magnitude induced by camera motion as well as the resulting geometric structure of CameraNoise. In our experiments, we found that $d=0.5$ provides a better balance between these factors and more faithfully represents the intended camera motion.
> > >
> > > In addition, in **Rebuttal-B**, we further show video generation results obtained by applying CameraNoise generated with $d=1000$ and $d=0.5$ to the video diffusion backbone across different scenes. As can be seen, when $d=1000$, the CameraNoise representation is already clearly degraded, and the generated videos fail to produce the desired camera motion control, especially for push-in and pull-out effects. By contrast, with $d=0.5$, the model is able to generate videos with the intended camera motion much more accurately. At the same time, we also observe that some sense of depth still remains in the generated videos when $d=1000$, suggesting that part of the depth plausibility comes from the learned generative prior of the diffusion model itself, rather than being explicitly provided by the degraded CameraNoise.
> > >
> > > Therefore, these results suggest that the main role of CameraNoise is to provide an explicit camera-motion prior for video diffusion. When this representation is overly simplified, for example, when $d$ becomes very large, the camera-motion prior carried by CameraNoise is significantly weakened and can no longer effectively guide the diffusion model to produce accurate camera control. Meanwhile, the diffusion model may still rely on its learned prior to recover a certain degree of scene depth and visual plausibility. In other words, the geometric plausibility of the final generated videos may partly come from the diffusion prior, but precise camera motion control still relies on the structured prior introduced by CameraNoise.
> > >
> > > Moreover, this also suggests that an overly simplified near-planar prior or a larger value of $d$ is insufficient for stable and accurate camera control, especially for camera motions such as push-in and pull-out, which are more sensitive to geometric representation.
> > >
> > > We thank the reviewer for this helpful suggestion, and we will incorporate these additional results into the revised version.

---

### Official Review · Reviewer_TdXg · 2026-03-13

**Soundness:** 3
**Presentation:** 4
**Significance:** 3
**Originality:** 4
**Overall Recommendation:** 5
**Confidence:** 2

**Summary:**

The paper proposes CameraNoise, a novel framework for camera-controllable video diffusion. To address the inaccuracy of direct numerical parameter injection and the appearance-entanglement issues of optical flow, the authors construct a Geometry-guided Reprojection Flow ($\mathcal{G}\_{r}$) from camera poses, smooth it via Lie algebra, and warp Gaussian noise using an advection PDE. This warped stochastic representation is then injected into the diffusion process to guide camera motion. Evaluated on datasets like RealEstate10K and DrivingDoJo, the method demonstrates strong performance in trajectory faithfulness, zero-shot generalization, and visual quality.

**Compliance With Llm Reviewing Policy:**

Affirmed.

**Final Justification:**

I appreciate the authors' rebuttal and will maintain my positive rating.

**Key Questions For Authors:**

(1) The Parallax Dilemma: How does the model generate realistic 3D parallax effects when the foundational GRFlow (Eq. 2) assumes a constant pseudo-depth $d$? Does the diffusion model's internal 3D prior (learned from vast video data) overwrite and compensate for this flat flow during the denoising process, or does the generated video occasionally exhibit 2.5D-like flat motion in highly complex scenes with extreme depth variations?

(2) Resolution Scaling: Could the authors provide a brief analysis or discussion on the computational complexity and memory footprint of the bipartite graph matching and PDE solving when scaling to much higher spatial resolutions?

(3) Ablation on the PDE Solver: Given the theoretical elegance of the PDE solver, I am curious about its absolute necessity compared to the GRFlow representation itself. If we were to replace the PDE-based CameraNoise generation with a simpler network (e.g., a lightweight convolutional encoder that directly injects GRFlow into the diffusion UNet/DiT), how much would the performance degrade? This would further highlight the indispensable value of the proposed PDE warping algorithm.

**Limitations:**

The authors have thoroughly discussed their method, but they should explicitly acknowledge the theoretical limitation of the constant pseudo-depth assumption ($d$) in Eq. (2) and its potential impact on generating videos with strong 3D parallax. A brief discussion on the scalability of the bipartite graph operations at higher resolutions would also complete the limitations section.

**Strengths And Weaknesses:**

Strengths:

(1) Rigorous and Elegant Mathematical Framework: Unlike many existing methods that treat camera conditioning as a black-box feature injection problem (e.g., passing Plücker coordinates through an MLP), this paper introduces a highly principled, geometry-grounded approach. Formulating the noise warping as an advection PDE solved via bipartite graph matching is mathematically elegant. It strictly preserves the Gaussian prior of the diffusion process while enforcing geometric consistency.

(2) Effective Disentanglement: By deriving the motion field strictly from intrinsic and extrinsic camera parameters (GRFlow), the method successfully decouples geometric camera motion from scene appearance. This effectively avoids the semantic conflicts and artifacts commonly seen in traditional optical-flow-based conditioning.

(3) Solid Empirical Performance: The method shows excellent quantitative and qualitative results, particularly in out-of-distribution (OOD) scenarios and cross-scene camera motion transfer. The ability to extract camera parameters from a source video and faithfully re-target them to a completely different generated scene is highly impressive.

Weaknesses:

(1) The Constant Pseudo-Depth Assumption (Parallax Limitation): While the mathematical derivation of GRFlow is elegant, Eq. (2) relies on a constant pseudo-depth $d$ to back-project 2D pixels into 3D space. This is a noticeable oversimplification of the 3D world, as it inherently ignores the 3D parallax effect (where foreground and background objects exhibit different displacement magnitudes under camera translation). Imposing a flat, constant-depth flow onto the noise space seems theoretically contradictory to realistic 3D scene dynamics, even if the neural network partially compensates for it.

(2) Scalability of the Graph-Based PDE Solver: Although the authors report an acceptable running time ($\sim$ 0.1s/frame) for the current resolution, solving the advection PDE via Jacobian-based density correction and bipartite graph matching inherently involves sparse matrix operations. It remains unclear how the memory footprint and computational cost of this mathematical machinery will scale when applied to much higher latent resolutions (e.g., generating 1080p or 4K videos).

(3) Reliance on Heuristic Tuning: The method still introduces several heuristic components, such as the dynamic scaling training (DST) hyperparameter $\eta$ and the empirical fusion coefficient $\lambda$, which might require non-trivial tuning when migrating the framework to other base diffusion architectures (e.g., from Wan 2.1 to Sora-like models).

---

> ### Author Rebuttal · Authors · 2026-03-31
>
> We thank Reviewer TdXg for the thoughtful and constructive comments. It is evident that the reviewer spent considerable time carefully reading our work and provided valuable insights and suggestions. We really appreciate the detailed feedback and the recognition of the strengths. Below, we address the specific points.
>
> > Q1: The Constant Pseudo-Depth Assumption.
>
> **Response:** Thank you for this insightful comment. In the proposed framework, the GRFlow is not intended to reconstruct exact 3D scene dynamics; instead, it serves as a camera-motion-specific geometric prior for noise warping. From this perspective, Eq. (2) is designed to capture the dominant global effect of relative camera motion while remaining scene-agnostic and free from appearance or object-motion leakage. Moreover, this approximation is empirically stable in practice. We tested different values of $d$ and found that a suitable constant value already generalizes well across nearly all scenes in our experiments. The CameraNoise examples shown in the second section of our project page were all obtained with the same fixed $d$, which further demonstrates the robustness of this design. We will clarify this point in the revision.
>
> > Q2: Scalability of the Graph-Based PDE Solver.
>
> **Table A**: Results of using the PDE solver.
>
> |Video Resolution|Latent Resolution|Time Cost (FPS)|Memory Usage (MB)|
> |------------------|-------------------|---------------|--------------|
> |1024×576|72×128|20.17|644.95|
> |1080p|135×240|13.15|764.38|
> |4K|270×480|5.60|1574.64|
>
> **Response:** Thank you for this insightful comment. Since diffusion models denoise in the latent space, and our video backbone model uses 8× spatial downsampling, the corresponding CameraNoise resolutions are only 240×135 for 1080p and 480×270 for 4K generation. As shown in Table A, on AMD EPYC CPU (64 cores), the PDE solver requires only 0.179 s/frame even for 4K. Moreover, the solver runs on the CPU and introduces no additional GPU memory overhead beyond the diffusion backbone. We believe this overhead is practical and will include this analysis in the revised version.
>
> > Q3: Reliance on Heuristic Tuning.
>
> **Table B**: Performance comparison of different models with the DST strategy.
>
> T/R/F=TransErr/RotErr/FVD
> |        | Wan-T | Wan-R | Wan-F  | Cog-T | Cog-R | Cog-F |
> |--------|------:|------:|-------:|------:|------:|------:|
> |$\eta$=0.8|0.158|0.284|191.20|0.176|0.312|224.80|
> |$\eta$=0.9|**0.155**|**0.277**|**188.36**|**0.171**|**0.304**|**217.40**|
> |$\eta$=1.0|0.166|0.302|197.50|0.184|0.331|233.60|
> |$\eta$=1.1|0.161|0.287|192.80|0.174|0.309|220.90|
>
> **Response:** Thanks for your suggestion. To verify the transferability of our DST strategy and fusion coefficient, we apply our CameraNoise-based controllable generation method to CogVideoX (Cog), another DiT-based backbone, and report a DST ablation in Table B. The results show that the proposed DST hyperparameter range generalizes well across backbones without requiring substantial tuning. Furthermore, the fusion coefficient is also stable across models, with values in [0, 0.1] consistently yielding good performance. We will also open-source our code and models in the future.
>
> > Q4: The Parallax Dilemma.
>
> **Response:** Thanks for your suggestion. CameraNoise is designed to encode camera motion, not the full depth variation of a scene. It provides a camera-aligned motion prior, while the final depth structure and parallax are further produced by the diffusion model during denoising, conditioned on the image and text inputs. Therefore, the final video is determined by the combination of a geometry-inspired camera prior and the model’s learned scene prior, rather than by the flat GRFlow alone.
>
> > Q5: Resolution Scaling.
>
> **Response:** Thanks for your suggestion. We give the detailed analyses in the rebuttal of Q2. These results demonstrate that our proposed method is efficient and can be readily adapted to high-resolution diffusion generation.
>
> > Q6: Ablation on the PDE Solver.
>
> **Response:** Thank you for pointing this out. Since this concern is closely related to a question raised by another reviewer, we refer the reviewer to our response to Q3 for Reviewer QUog. There, we compare several conditioning strategies built on the same base model, including value injection, GRFlow injection, CameraNoise injection, and a hybrid strategy combining CameraNoise with value injection. For GRFlow injection, we use a conditioning-like design that concatenates the corresponding tokens before the DiT backbone. The results show that, although GRFlow injection provides some controllability, its high-dimensional temporal representation is more difficult for the diffusion model to fit, leading to degraded performance. In contrast, CameraNoise achieves stronger overall performance and controllability than both direct value injection and hybrid injection. This is a really helpful suggestion, and we will include these results in the revised version.

---

> > ### Author Rebuttal · Reviewer_TdXg · 2026-04-02
> >
> > Thank you for the rebuttal; my concerns are addressed and I maintain my positive score.

---

### Official Review · Reviewer_QUog · 2026-03-14

**Soundness:** 3
**Presentation:** 4
**Significance:** 3
**Originality:** 2
**Overall Recommendation:** 4
**Confidence:** 5

**Summary:**

This paper studies camera-controllable video diffusion and argues that direct camera-feature injection is often too weak for faithful trajectory control. The work proposes to address a central area in controllable video generation, namely, how to better preserve geometry and motion consistency under camera changes. The authors seek to present a fundamental challenge: numerical camera embeddings do not map cleanly to visual structure, which can lead to distortions and inaccurate motion. To tackle this, the paper proposes CameraNoise, which injects camera motion into the diffusion noise space via an appearance-agnostic GRFlow and a PDE-based warping scheme. Experiments on multiple datasets suggest improved camera faithfulness and competitive visual quality over prior methods.

**Compliance With Llm Reviewing Policy:**

Affirmed.

**Final Justification:**

Thanks to the authors for their rebuttal, which addressed some of my concerns. However, several of the remaining concerns were still not adequately resolved. In particular, the discussion and comparison with the most closely related prior works: *How I Warped Your Noise: A Temporally-Correlated Noise Prior for Diffusion Models* and *Go-with-the-Flow: Motion-Controllable Video Diffusion Models Using Real-Time Warped Noise*, remain insufficient. From the rebuttal, the main distinction I can identify is in the choice of representation. Yet the proposed GRFlow appears to model only one specific form of motion, namely camera motion, whereas optical-flow-based methods can capture a broader class of motions, including camera motion, suggesting that the underlying problem formulation is still quite similar.

Moreover, it remains unclear why **Ours-Combine** underperforms **Ours-CameraNoise** in Table A. The authors did not provide meaningful discussion or analysis on this point; reporting the numbers alone is not enough to convincingly explain the behavior. In addition, the response to my key question felt somewhat shallow, and some parts of my question were left unaddressed. As a result, I was not able to draw substantially more insight from the rebuttal. For these reasons, I will keep my original score.

**Key Questions For Authors:**

How do the authors view the two types of camera condition manners (noise warping-based and injection-based) in a practical perspective? Are they showing a clear difference during the applications? How to choose these manners based on different requirements, for example, under different scenarios, movement, and domains, which one is expected to perform better? From my experience, some of them show very few differences in the final generation performance if the model is trained on enough data and resources. Even using the raw values of the camera parameters can achieve promising results. So, would the complex and hand-crafted camera condition manners be a bottleneck for scaling up the world model? I believe a work that can unify these manners, evaluate them using the same architecture, training data, and benchmarks, and provide some insightful discussions on top of them is much more valuable than designing how to use a "new" representation.

**Limitations:**

Please refer to the weaknesses part.

**Strengths And Weaknesses:**

Strengths:

+ The paper does not just add another camera encoder; instead, it reframes camera control as structured noise construction. That is a well-motivated conceptual shift, and it directly targets a real weakness of prior methods that only inject camera features into intermediate activations.
+ The experiments are reasonably extensive, including T2V, I2V, zero-shot transfer to other datasets, qualitative examples, and ablations on smoothing, fusion ratio, and DST. This gives the paper a stronger experimental backbone than many papers in this space, where one often sees only cherry-picked videos

Weaknesses:
- The discussion and comparison with the most related previous methods ("How I warped your noise: a temporally-correlated noise prior for diffusion models" and "Go-with-the-flow: Motion-controllable video diffusion models using real-time warped noise") are insufficient. The key concept in this work, GRFlow, is essentially the same as the optical flow representation used in the above two methods. Both of them leverage the pixel-wise displacement field; the only difference seems to be that GRFlow describes the global movement of the camera, and optical flow describes the local movement of the objects. They generally represent the motion in the same space.
- Some related camera-controllable generation methods are missing in the related works and introduction. For example, "Bolt3D: Generating 3D Scenes in Seconds", "Thinking with Camera: A Unified Multimodal Model for Camera-Centric Understanding and Generation", "Stable Virtual Camera: Generative View Synthesis with Diffusion Models", "Cameras as Relative Positional Encoding", etc. The authors are suggested to provide some discussions on these representation-based camera condition designs and highlight their uniqueness.
- The ablation study of using the same network architecture with different camera conditions is missing. It seems the authors only claim the noise warping manner performs better than the classical "injection" way. However, it's still unclear why and how this work performs better.  Would it bring further improvement if we combine both methods within a single model?
- I believe there should be some limitations on the noise warping manner, especially compared to the classical "injection" way, but the authors failed to discuss them. One potential limitation could be the error propagation across the warped noises for the final video generation. The computational complexity could be another limitation. More discussions are expected to be provided.
- Some important implementation details, such as the computation resources (both training and inference), are lacking in the experiment part.

---

> ### Author Rebuttal · Authors · 2026-03-31
>
> We thank Reviewer QUog for the thoughtful and constructive comments. It is evident that the reviewer spent considerable time carefully reading our work and provided valuable insights and suggestions. We really appreciate the detailed feedback and the recognition of the strengths. Below, we address the specific points.
>
> > Q1: The only difference seems to be that GRFlow describes the global movement of the camera, and optical flow describes the local movement of the objects.
>
> **Response:** We apologize for any confusion. The fundamental difference lies in what the field represents and what it introduces into diffusion.
>
> - Representation: Optical-flow-based methods model image motion and thus entangle camera motion, object motion, and appearance. In contrast, our GRFlow models only camera-induced geometric reprojection from camera parameters, without injecting shape into the noise prior.
> - Effect on diffusion: As shown in Fig. 1 (T2V), optical-flow-based methods may leak object-specific shape priors into denoising and cause artifacts, while GRFlow avoids such shape leakage and provides a cleaner camera control signal.
>
> > Q2: Provide some discussions on these representation-based camera condition designs and highlight their uniqueness.
>
> **Response:** Thanks for the suggestion. These works are closely related to ours, covering single-image 3D scene generation, text-driven camera rotation, and multi-view generation. They provide complementary background for our study. We will cite and discuss them in the Introduction and Related Work of the revised version.
>
> > Q3: It's still unclear why and how this work performs better (than the injection way). Would it bring further improvement if we combine both methods within a single model?
>
> **Table A:** Comparison of different methods on MultiCamVideo100 (MC) and DrivingDojo100 (DD). Lower is better.
>
> T/R/F=TransErr/RotErr/FVD
>
> |M|MC-T|MC-R|MC-F|DD-T|DD-R|DD-F|
> |----------------------|-----:|-----:|------:|-----:|-----:|------:|
> |Wan2.1-Fun|0.245|0.238|456.32|0.842|0.294|531.32|
> |Ours-value-injection|0.228|0.311|400.34|0.973|0.289|374.12|
> |Ours-GRFlow-injection|0.262|0.336|438.57|1.041|0.318|418.46|
> |Ours-combine|0.195|0.234|367.81|0.421|0.268|248.93|
> |**Ours-CameraNoise**|**0.191**|**0.230**|**362.57**|**0.397**|**0.265**|**242.67**|
>
> **Response:** Thank you for pointing this out! We conducted additional ablations using the same base model, Wan 2.1. We trained several variants on RealEstate10K and DynPose-100K, including a CameraCtrl-style value injection variant, a direct GRFlow injection variant, and a variant that combines value injection with CameraNoise. We also evaluated Wan2.1-Fun 14B, an open-source method with a similar value-injection design. We then performed zero-shot I2V evaluations on two benchmarks. The results show that CameraNoise achieves the best controllability and generalization. We believe that geometry-guided control in the noise space provides a more powerful and principled approach. In contrast, combining CameraNoise with value-based fusion degrades the performance. This is a real helpful suggestion, and we will include these results in the revised version.
>
> > Q4: The error propagation across the warped noises for the final video generation. The computational complexity could be another limitation. More discussions are expected to be provided.
>
> **Response:** Thank you for the suggestion. Error propagation from CameraNoise is not a major issue during denoising, consistent with prior warped-noise methods such as Go-with-the-Flow. A practical limitation lies in the quality of estimated camera poses without GTs, which may affect CameraNoise in challenging cases such as severe camera shake or shot transitions. In terms of efficiency, please refer to Appendix G and the rebuttal Table A in Q2 of our response to Reviewer TdXg for more details. Thanks!
>
> > Q5: Some important implementation details, such as the computation resources (both training and inference), are lacking.
>
> **Response:** Thank you for pointing this out. Due to the page limit, we included the runtime details for GRFlow, CameraNoise, training, and inference in Appendix G, where the computational cost is reported explicitly.
>
> > Q6: Key Questions: the difference between injection and CameraNoise, trained on enough data and resources, applied to the world model.
>
> **Response:** Thank you for this thoughtful comment. We agree that raw parameters may also work well with sufficient data and resources. However, Q3 response shows that, even under the same training setup, different conditioning designs lead to different performance. We believe CameraNoise provides a more structured, diffusion-aligned inductive bias, which improves learning efficiency and controllability. It also adds little extra overhead in practice. Your comment is very inspiring, and we will further explore CameraNoise, a model-agnostic representation in world-model-related settings. We will also open-source our code and models.

---

> > ### Author Rebuttal · Reviewer_QUog · 2026-04-03
> >
> > Thanks to the authors for their rebuttal, which addressed some of my concerns. However, several of the remaining concerns were still not adequately resolved. In particular, the discussion and comparison with the most closely related prior works: *How I Warped Your Noise: A Temporally-Correlated Noise Prior for Diffusion Models* and *Go-with-the-Flow: Motion-Controllable Video Diffusion Models Using Real-Time Warped Noise*, remain insufficient. From the rebuttal, the main distinction I can identify is in the choice of representation. Yet the proposed GRFlow appears to model only one specific form of motion, namely camera motion, whereas optical-flow-based methods can capture a broader class of motions, including camera motion, suggesting that the underlying problem formulation is still quite similar.
> >
> > Moreover, it remains unclear why **Ours-Combine** underperforms **Ours-CameraNoise** in Table A. The authors did not provide meaningful discussion or analysis on this point; reporting the numbers alone is not enough to convincingly explain the behavior. In addition, the response to my key question felt somewhat shallow, and some parts of my question were left unaddressed. As a result, I was not able to draw substantially more insight from the rebuttal. For these reasons, I will keep my original score.

---

> > > ### Author Response · Authors · 2026-04-04
> > >
> > > We sincerely thank Reviewer QUog for the continued thoughtful feedback and for the constructive discussion. We will further clarify the relation to previous methods and provide a more explicit discussion of the behavior of “Ours-combine” in Table A in the revised version.
> > >
> > > Regarding the relation to prior warped-noise methods, we agree that the formulations are related at a high level, since both introduce temporally structured priors into the diffusion process. However, our goal is not to model the most general class of motion fields, but to provide **a camera-centric geometric prior** specifically for camera-controllable video generation. In this sense, the difference is not only the representation itself, but also the control target and inductive bias. Optical-flow-based representations are more general, but they also entangle camera motion with object motion, non-rigid deformation, appearance variation, and flow estimation noise. By contrast, GRFlow is designed to isolate the motion induced by camera transformation only. We intentionally adopt this more restricted formulation because, for camera control, a cleaner and more geometry-aligned signal is often preferable to a more general but entangled motion representation.
> > >
> > > This distinction also helps explain why Ours-Combine underperforms Ours-CameraNoise in Table A. CameraNoise already injects a clean camera prior at the noise initialization stage, which is well aligned with the denoising process. When additional value-based control signals are fused into the backbone, they are not necessarily complementary to CameraNoise. In our experiments, the value-based control itself is weaker than CameraNoise in terms of camera controllability, so the fused model may rely on a less effective and more entangled signal. As a result, this additional branch can weaken the stable control effect provided by CameraNoise, rather than strengthen it.
> > >
> > > We will expand the discussion of the differences from optical-flow-based warped-noise methods in the revised version, and also add a more explicit analysis of why the combined variant underperforms the CameraNoise-only design. We thank the reviewer again for highlighting these important points.

---

### Decision · Program_Chairs · 2026-04-30

**Decision:**

Accept (regular)

**Comment:**

All 4 reviewers recommend either Accept or Weak Accept after considering the rebuttals. The AC has reviewed the rebuttals/comments and agree with the reviewers’ consensus to accept this paper. This paper is strong in its theoretical elegance, a principled approach to address artifacts caused by flow-based conditioning, and strong empirical results.

The authors are encouraged to incorporate the rebuttal to enhance the final version, including acknowledging the theoretical limitations of the pseudo-depth assumption, its potential impact on 3D parallax, comparisons with optical-flow-based noise warping methods, and performance on dynamic scenes.